# LVstyler: LoRA-enhanced Varied High-Quality Texture Generation with Text Alignment

## Abstract

We introduce LVstyler, an innovative generative framework that extends beyond the conventional 0-to-1 UV texture synthesis, specializing in a more varied style transfer for 3D meshes while preserving geometric fidelity. The core challenge lies in maintaining style consistency across complex 3D surfaces without introducing style-agnostic artifacts. Other methods leverage a pre-trained texture generation model, which primarily relies on a diffusion model, producing an initial texture map. However, due to the limited styles that a simple pre-trained diffusion-based model can generate for objects, these methods can only handle short and object-based prompts, rather than styling prompts. Therefore, we integrate an optimization-based texture generation model in the image space, specifically modifying it with two LoRA (Low-Rank Adaptation) extensions for shape consistency and UV map space adaptation. Through this technique, LVstyler can produce varied high-quality UV textures that allow more imagination through the detailed styling text guidance, significantly advancing the state-of-the-art in texturing 3D objects.

## 1 Introduction

Recent advancements in generative AI have significantly transformed 3D synthesis, enabling the creation of novel 3D content. A pivotal development in this field is neural radiance fields (Mildenhall et al., 2020), which employ neural networks to implicitly learn mathematical representations. Subsequent innovations on distillation-based models (Poole et al., 2022; Chen et al., 2023b; Wang et al., 2023b; Yi et al., 2023; Tang et al., 2023) mainly learn from well-trained diffusion models and further enhance 3D reconstruction quality. The introduction of Gaussian splatting (Kerbl et al., 2023) marks a significant milestone due to its faster iteration and reduced computational cost, leading to its widespread adoption in AI 3D generation pipelines. Among all 3D representations, the mesh representation is the most common in the industry. It is often combined with UV textures, which involve unwrapping a 3D model's surface into a 2D representation in the UV space, where U and V are the horizontal and vertical axes of the texture map, respectively. However, the generated meshes, typically observed with flawed and incomplete textures, are often the obstacles to applying the automatically generated 3D content into the realistic workflow. Advancements in texture synthesis are an emerging topic to be solved under this limitation. Within practical 3D content pipelines, UV texture stylization therefore functions as a critical bridge between initial asset generation and downstream editing: game studios frequently require multiple visual skins for the same character or prop, product designers explore style variants over a fixed geometry, and animation or film production relies on stylized assets that share common mesh topology. Yet most existing research targets either 0-to-1 3D generation or high-fidelity editing, leaving this intermediate re-stylization stage comparatively underexplored despite its central role in production workflows. Current methods (Richardson et al., 2023; Chen et al., 2023a; Liu et al., 2024b) mainly rely on diffusion models to generate and paint semantically coherent UV textures. Furthermore, by focusing on lighting problems that affect texture realism, some research efforts (Zeng et al., 2023; Zhang et al., 2024) aim to provide texture solutions that maintain quality under varied lighting conditions. Meanwhile, other approaches (Bensadoun et al., 2024; Chen et al., 2024) prioritize speed in texture generation, enabling faster iterations and reducing computational cost.

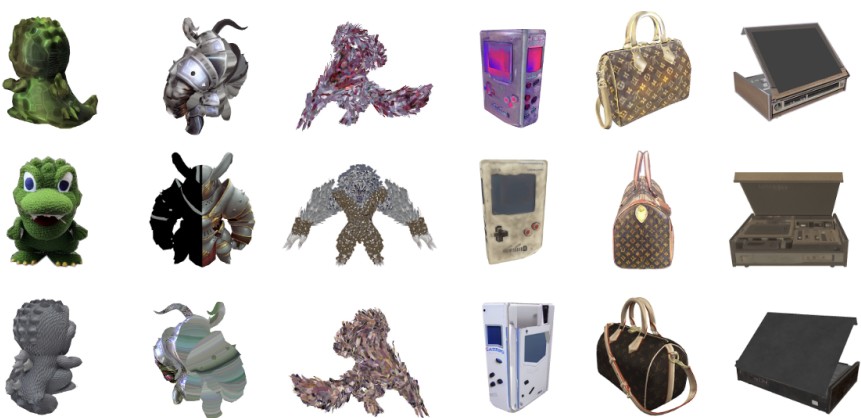

Figure 1: A gallery of generated texture results by LVstyler. Our method disentangles shape and UV information and generates varied and detailed textures for 3D objects.

However, texture synthesis is actually a dual task: achieving faithful content reproduction and incorporating artistic styling. Despite recent advancements, diffusion models struggle with style transfer and artistic rendering, as they are trained to generate content-consistent textures using millions of in-domain images. Infusing stylistic nuances that diverge from training data proves challenging, demanding inherent creativity and flexibility. Models must preserve 3D structural integrity while adapting dynamically to abstract styles, enhancing adaptability for diverse creative applications. This challenge holds profound implications for real-world applications in product design, game content, film animation, and customizable aesthetics.

Therefore, to disentangle these dual needs, we develop a two-stage LoRA-enhanced framework, LVstyler, incorporating both object and styling prompt guidance. We utilize existing texture inpainting models that generate multi-view images via pre-trained 2D diffusion models and project them onto 3D mesh surfaces for initial texture maps. LVstyler then refines these textures through LoRA-enhanced styling models designed for shape-consistent generation and UV space adaptation. Importantly, our method does not presuppose clean, fixed UV mapping. It accepts UV maps from various sources, including AI-generated textures and Human-defined UV maps, which extend the flexibility of our model. Our model directly stylizes UV maps, independent of diffusion networks and back-projection schemes, facilitating integration into industry-grade pipelines while significantly reducing training costs and computational requirements compared to diffusion-based approaches. Our contributions are summarized as follows:

- We present a novel pipeline that is substantially different from current diffusion-based approaches. It offers creative freedom on top of original, simple inpainting models, while it can be generalized to stylize any existing well-defined multiple UV maps.

- We integrate two LoRA modules within image styling networks to ensure the spatial preservation of the UV map and styling details. This approach introduces a style adaptation directly in the UV space, addressing the challenge of lacking a stylized UV map texture dataset.

- We innovate the hybrid training schema of online learning and modular pretraining. This schema keeps continuous learning capability while allowing the model to have the pretraining modules to adapt to the new information that can be disentangled.

- We collect an extensive dataset, LVstyle, compiling raw, masked, and styled UV maps. This dataset will be made publicly accessible and serve as a substantial asset for the research community in 3D styling tasks.

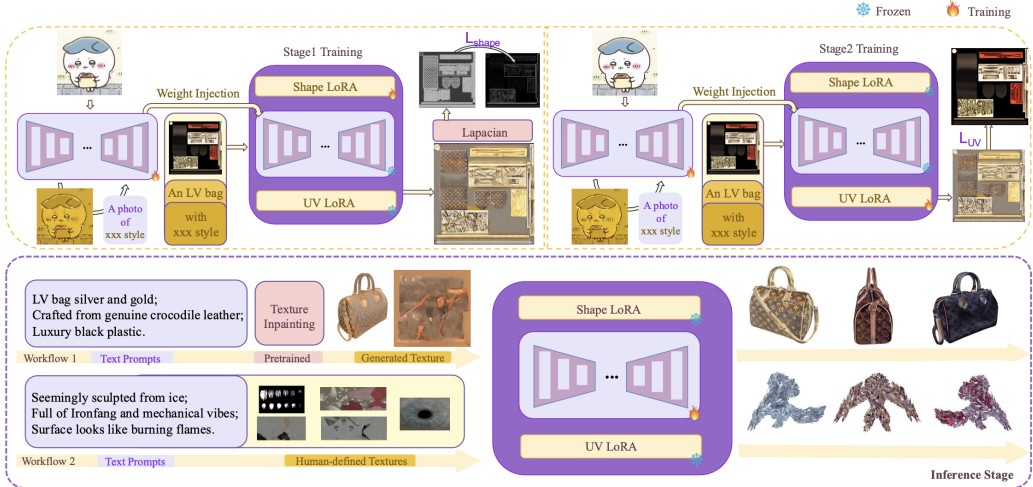

Figure 2: Flowchart of the LVstyler framework. In training stage 1 (top left), we train an image space styling (ISS) model on an arbitrary image, optimizing for a style guided by CLIP, and inject the weights into LVstyler. We then train the shape LoRA module, keeping ISS and UV LoRA weights frozen. Similarly, in stage 2 (top right), we train the UV LoRA for adapting the style transfer in the UV space. The bottom panel depicts the inference workflow, where LVstyler uses generated UV textures or multiple UV maps, alongside a text prompt, to create varied, detail-rich UV textures for 3D objects.

## 2 RELATED WORK

**Image Style Transfer**. Recently, style transfer evolves significantly, starting with the foundational work by framing the process as an iterative pixel optimization task (Gatys et al., 2016). This framework is further converted into real-time feed-forward nets (Johnson et al., 2016; Ulyanov et al., 2016). Initial trials to achieve arbitrary style learning on various statistical matching techniques (Li et al., 2017; Huang & Belongie, 2017; Li et al., 2019) focus on aligning style statistics such as covariances, mean–variance. Further refinements through attention mechanisms (Park et al., 2019; Yan et al., 2019; Ding et al., 2021) and optimization techniques (Huang et al., 2021) improve the learning of complex, non-linear style features and enhance spatial fidelity and realism. The language guidance represents another stream of research. StyleCLIP (Patashnik et al., 2021) and StyleGAN-NADA (Gal et al., 2022) utilize CLIP's textual capabilities to guide StyleGAN, whereas CLIPstyler (Kwon & Ye, 2022) offers direct image stylization without relying on GAN priors. As style transfer techniques become mature, the emphasis shifts towards offering fine-grained control. For instance, learnable regions (Abdal et al., 2022) employ text-aligned bounding boxes to specifically redraw user-indicated areas, while CLIPAG (Ryu et al., 2023) fine-tunes CLIP to generate perceptually aligned gradients that independently drive text-to-image synthesis. Recently, diffusion models have produced high-resolution, controllable styling results with innovations such as training-free style injection in diffusion (Kim et al., 2023), mask-aware DiffStyler (Matsumori et al., 2023), InstantStyle-Plus (Huo et al., 2024), and transformer-based U-StyDiT (Wang et al., 2024). Yet, these operations remain in the 2D image space. The true 3D texture adaptation and view-consistent stylization are still unachievable.

**Styling for 3D Objects**. Early work on text-driven 3D styling applies CLIP supervision to predefined human meshes, such as MeshStyle (Hong et al., 2022) and StyleAvatar (Jetchev, 2021). Subsequent methods further improve this idea. Text2Mesh (Michel et al., 2022) introduces lightweight per-vertex edits, while CLIP-Mesh (Khalid et al., 2022) unifies geometry, texture, and normals through differentiable rendering. Recent systems boost visual fidelity by harnessing 2D diffusion models. TEXTure (Richardson et al., 2023), TexFusion (Mabsout et al., 2023) and Text2Tex (Reich et al., 2023) iteratively generate view-consistent UV maps, while generative or hybrid pipelines like Texturify (Yuan et al., 2023), Mesh2Tex (Zhang et al., 2023c), DreamFusion (Poole et al., 2022) and Latent-NeRF (Li et al., 2023) synthesize shape and appearance simultaneously, while they often struggle to disentangle style from lighting and geometry. Meanwhile, Paint3D (Zeng et al., 2023) achieves lighting-agnostic 2K textures by combining depth-conditioned multi-view generation with

UV-space inpainting and high-frequency de-lighting diffusion. More recent approaches focus on multi-view consistency and scalability. SyncMVD (Liu et al., 2024b) synchronizes multi-view diffusion processes to achieve coherent texturing across different viewpoints, while TEXGen (Yu et al., 2024b) proposes a generative diffusion model operating directly on mesh textures with attention-guided sampling. GenesisTex (Yu et al., 2024a) adapts a 700M-parameter image denoiser directly to the UV space for feed-forward texture synthesis. Hunyuan3D-2.0 (Team, 2025) scales diffusion models for high-resolution textured 3D asset generation, achieving impressive visual quality but with substantial computational requirements. However, most existing models rely on diffusion-based inpainting with a large number of parameters, leading to high computational cost.

**LoRA Fine-tuning**. Low-rank adaptation (Hu et al., 2022) inserts compact rank-decomposition matrices into frozen pre-trained weights, allowing parameter-efficient fine-tuning while drawing on earlier insights into collaborative discourse and planning strategies (Grosz & Kraus, 1996; Kautz & Selman, 1992). Subsequent work has advanced LoRA along a continuum of memory frugality, rank flexibility, context length, and modality. QLoRA (Dettmers et al., 2023) combines LoRA with 4-bit quantization to fit 65B-parameter language models onto a single GPU. AdaLoRA and ALoRA (Zhang et al., 2023b; Liu et al., 2024a) dynamically prune or re-allocate ranks across layers to maximize parameter-budget utility. LoRA+ (Wang et al., 2023a) accelerates convergence by decoupling learning rates for the two update matrices, while LongLoRA (Zhang et al., 2023a) extends context windows to 32–100k tokens by jointly tuning sparse attention and positional embeddings. More recently, ElaLoRA (Kong et al., 2024) elastically grows or shrinks ranks on-the-fly to meet tight memory constraints. In diffusion models, cross-modal extensions of LoRA quickly follow. LoRA is now the de facto lightweight tuner for stable diffusion (Hugging Face Team, 2023), while DiffLoRA (Wu et al., 2024) treats a latent-diffusion hyper-network as a LoRA weight generator, enabling zero-shot personalization without test-time optimization. Collectively, these variants cement LoRA as a leading adaptation framework for ever-larger and multimodal models, although fully geometry-aware 3D texture fine-tuning and view-consistent stylization remain as open challenges.

# 3 METHODOLOGY

## 3.1 PROBLEM FORMULATION

As discussed in Section 1, texture synthesis is actually a dual task—achieving photorealism and incorporating artistic styling. While the current methods focus more on inpainting UV texture to achieve photorealism, our work is dedicated to solving the artistic styling for 3D objects, allowing a single 3D object with varied styling. This means that, except for keeping the original details and structure, the model should achieve a new injection style.

A fundamental challenge in UV texture stylization is the absence of large-scale datasets containing pixel-aligned stylized UV maps. To circumvent this limitation, we propose a three-part decoupled supervision framework that obviates the need for explicit ground-truth stylized textures: (1) *Style supervision* is achieved through CLIP-based contrastive learning, which enforces semantic alignment between the generated texture and the target style description; (2) *Structural supervision* leverages edge detection operators applied to original UV maps, providing geometric constraints that preserve object boundaries and surface details; and (3) *Domain supervision* employs structural alignment between stylized outputs and original UV maps, ensuring that the stylization process maintains the spatial coherence inherent to UV texture representations. This multi-objective design fundamentally mitigates over-reliance on CLIP as the sole supervision signal—unlike methods that depend purely on CLIP guidance, our framework combines vision-language alignment with geometry-aware constraints and domain-specific regularization, enabling robust stylization even for prompts where CLIP's text-image alignment may be imprecise.

We aim to minimize the distance function for the styled texture and prompt, $F$, under a regularization function $G$ that is the distance function for the styled texture and the original image. Thus, the problem should be formulated as follows,

$$\min F(Y_{\text{styled}}, T_{\text{styling}}) + \lambda \cdot G(Y_{\text{styled}}, Y_{\text{original}}),$$

where $Y_{\text{styled}}$ is the styled output, $T_{\text{styling}}$ is the styling prompt, and $Y_{\text{original}}$ is the original object details.

Following this route, we propose LVstyler, an innovative pipeline to disentangle the style, shape, and UV texture space, allowing the model to adopt new styles to UV maps while preserving original shape details. Our model can be divided into two parts. The first part is a base styling model inspired by CLIPstyler (Kwon & Ye, 2022). It utilizes the online learning training schema to optimize the style learning under the supervision of CLIP in the image space, $I \in \mathbb{R}^{n \times n \times 3}$. In the second part, we inject the optimized weights from the base model and support the shape consistency and UV space adaptation with two LoRA modules, where the UV space is denoted as $R \in \mathbb{R}^{n \times n \times 3}$. Corresponding to our three-part supervision framework, we adopt a two-stage training strategy to properly disentangle these objectives: Stage 1 trains the shape LoRA to enforce structural supervision, while Stage 2 trains the UV LoRA for domain supervision. Style supervision through CLIP operates continuously across both stages. This sequential approach prevents conflicting gradients that would arise from joint optimization—shape preservation demands high-weight edge constraints whereas UV adaptation requires relaxed content fidelity to project styles effectively. By freezing earlier modules before training subsequent ones, each component learns its designated role without interference, yielding stable optimization dynamics and superior final quality.

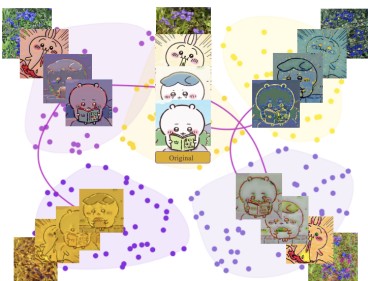

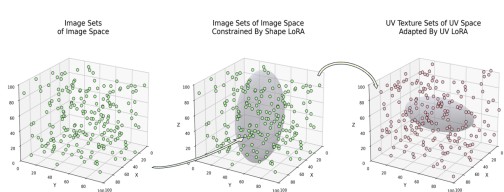

(a) Illustration of image space style learning.

(b) Styling simulation

Figure 3: Methodology illustrations: (a) image space style learning process. The purple line mimics the ISS training, which can be regarded as the model searching in the parameter space to learn a certain style. (b) UV styling simulation. The left graph simulates styled images in the image space, usually with a diverse range of choices. The middle part represents the shape LoRA constraining the space to ensure structural similarity. The projection between the image space and UV space is shown in the second and third graphs.

## 3.2 IMAGE SPACE STYLING ONLINE LEARNING

During training, our model takes an arbitrary photo as the subject of image space styling. The purpose of this stage is to adapt the CLIP model to supervise our training. Originally, CLIP is trained on image and text pairs. Directly feeding the UV map confuses CLIP supervision, which is not reasonable for the training schema. Meanwhile, due to the simplicity of the CNN encoder-decoder model in CLIPstyler, the optimization-based approach can quickly iterate and adapt to any new style, which should be superior to diffusion families that require fine-tuning on massive styled data. Thus, we input the image in the image space to optimize weight $W$ in the network ISS (image space styling), so that any generated figure from the network is closely related to the style given in the text prompt. The objective function can be formulated as

$$Y_{\text{styled}} = \text{ISS}(Y_{\text{original}}, T_{\text{styling}}),$$

and thus the learned parameters are

$$W^*_{\text{styled}} = \arg \min_W F(Y_{\text{styled}}, T_{\text{styling}}).$$

To achieve the minimization of the objective, we adopt the loss function utilized in CLIPstyler for style learning, comprising four distinct components. We use directional CLIP loss $L_{\text{dir}}$ to modify global content image aspects like color tone and semantics, PatchCLIP loss $L_{\text{patch}}$ for local texture adaptation, content loss $L_c$ to preserve input image content, and total variation regularization loss $L_{\text{tv}}$. Consequently, the total loss function $L_{\text{ISS}}$ is formulated as

$$L_{\text{ISS}} = \lambda_{\text{dir}} L_{\text{dir}} + \lambda_{\text{patch}} L_{\text{patch}} + \lambda_c L_c + \lambda_{\text{tv}} L_{\text{tv}},$$

where $\lambda_{\text{dir}}$, $\lambda_{\text{patch}}$, $\lambda_c$ and $\lambda_{\text{tv}}$ are tuning parameters.

### 3.3 UV Space Styling Learning

LoRA imposes an assumption that when adapting pre-trained models, the necessary weight adjustments exhibit a low intrinsic rank. For a pre-trained weight matrix $W$ with dimensions $d \times k$, the update can be expressed as a low-rank decomposition $W + \Delta W = W + B \cdot A$, where $B \in \mathbb{R}^{d \times r}$, $A \in \mathbb{R}^{r \times k}$, with $r \ll \min(d, k)$. With this technique, we could learn pretrained and incremental knowledge to minimize the distance function $G(\cdot)$ for $Y_{\text{styled}}$ and $Y_{\text{original}}$.

After the iterative optimization of the ISS model, we inject the weights into our LVNet, which is composed of the ISS network, shape LoRA (SL), and UV LoRA (UL). The injection process can be expressed as

$$W_{\text{ISS}} = W_{\text{styled}}^*,$$
$$W_{\text{LVNet}} = W_{\text{ISS}} + \Delta W_{\text{SL}} + \Delta W_{\text{UL}}.$$

Stage 1 trains the shape LoRA. Instead of demanding a full shape identical output, clear element separation in the UV texture is needed, not only for preserving details on 3D objects but also to allow industry practitioners to further modify it. Thus, different from normal loss calculation directly on the output and original data, we further process the output with a Laplacian kernel to extract the edge masks $M_{\text{styled}}$ and $M_{\text{original}}$ from both the styled texture $Y_{\text{styled}}$ and original UV texture $Y_{\text{original}}$, for the shape loss calculation. We define the loss function as

$$L_{\text{stage1}} = L_{\text{ISS}} + L_{\text{shape}},$$

where

$$L_{\text{shape}} = \sum_{i,j} |M_{\text{styled}}(i,j) - M_{\text{original}}(i,j)|.$$

Stage 2 involves the training of the UV LoRA module, where the ISS and shape LoRA are fixed. Since we have learned the style in the realm of image space, the most important thing is to learn how to project the style onto the UV space. We freeze the $W_{\text{ISS}}$ and $\Delta W_{\text{SL}}$ while optimizing $\Delta W_{\text{UL}}$ for learning incremental information on the projection of style. This minimizes the loss between the generated styled texture $Y_{\text{styled}}$ and the original texture. The loss function is designed to align the distribution of generated UV textures with the original distribution as follows,

$$L_{\text{stage2}} = L_{\text{ISS}} + L_{\text{UV}},$$

where

$$L_{\text{UV}} = \sum_{i,j} |Y_{\text{styled}}(i,j) - Y_{\text{original}}(i,j)|.$$

### 3.4 Hybrid schema: online learning with offline modular constraints

We propose a novel inference schema that combines online test-time optimization with freezing of offline-trained modules. Concretely, the Shape LoRA (trained in Stage 1) and the UV LoRA (trained in Stage 2) are frozen at inference and act as distributional constraints, while the ISS backbone remains unfrozen and is optimized per prompt under CLIP-based text alignment. This hybrid inference strategy addresses the fundamental challenge of balancing content preservation with style adaptation through selective parameter optimization. The LoRA modules are designed to learn gradual data distribution shifts from natural images to UV space, contrasting with conventional approaches that fine-tune diffusion models for specific character distributions. Freezing these LoRA modules provides distributional constraints that guide transformation pathways, whereas the primary styling effects emerge from the unfrozen ISS backbone during online optimization. This approach enables stable optimization dynamics by decoupling offline structural/stylistic learning from online adaptation. The modular design also facilitates efficient adaptation to diverse styling requirements without catastrophic forgetting of geometric constraints.

## 4 Experiments

### 4.1 Implementation Details

**Data** The study uses textured meshes from the Objaverse dataset (Deitke et al., 2022) and Model-Net40 (Wu et al., 2015), a collection of CAD point cloud models from 40 categories. The dataset is

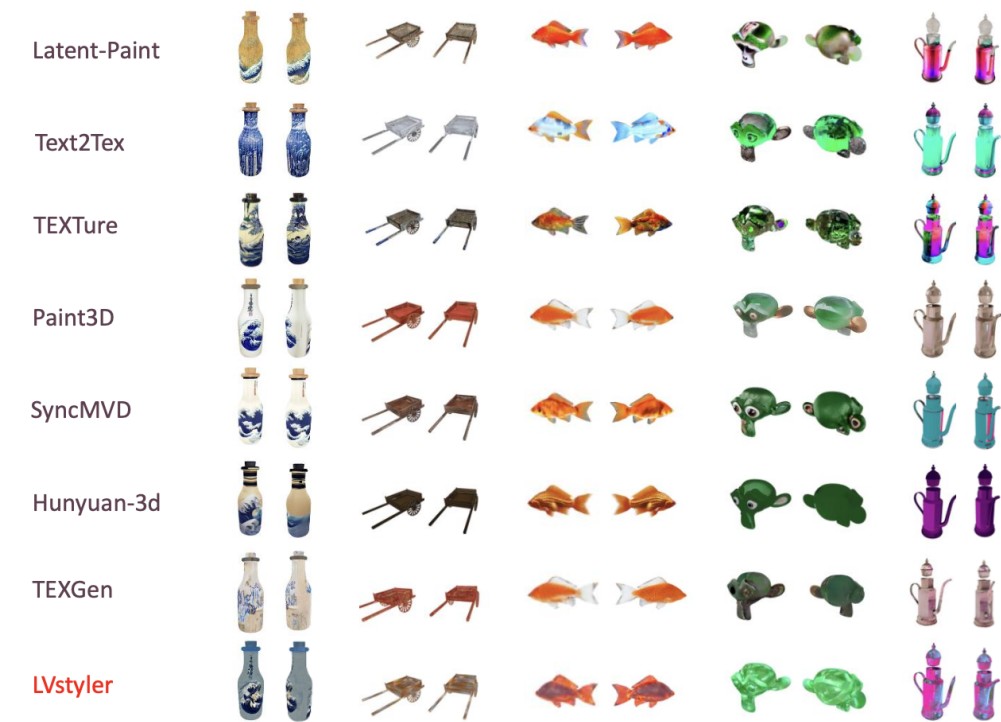

Figure 4: Qualitative comparisons on texture generation from text prompts. The figure compares Latent-NeRF (row 1), Text2Tex (row 2), TEXTure (row 3), Paint3D (row 4), SyncMVD (row 5), Hunyuan3D-2.0 (row 6), TEXGen (row 7), and **LVstyler (row 8)**. Each object is displayed from two viewpoints to provide multiview comparison across different models. From left to right, the prompts are: a decorative bottle with the Great Wave off Kanagawa by Hokusai, a trolley cart with metal engraving, a goldfish with fire skin, a green crystal turtle, and a steaming kettle in a sleek neon style featuring vivid pinks, electric blues, and radiant greens. The video demonstrations for 360 degree comparison are attached in Appendix A.7

preprocessed, resulting in 410 object textures from Objaverse, 120 from ModelNet40, and 140 synthetic textures. For each texture, masking is performed to extract the edges and noise details of UV maps, and image-styling models are applied to produce stylized images, constituting our LVstyler dataset. More details on preprocessing and dataset collection are given in Appendix A.4.

**Experimental settings** Our proposed model only consists of a total of 615.27K parameters, which is designed to balance quality and efficiency. During the training phase, 16 RTX 4090 GPUs or 8 NVIDIA A800 GPUs take approximately 10 hours to complete 200K steps of stage 1 training and 5 hours for 100K steps of stage 2 training. For inference, the model can run efficiently on a single RTX 4090 GPU in about 1 minute.

**Evaluation Metrics** For our quantitative analysis, we employ metrics such as Fréchet inception distance (FID) (Heusel et al., 2017), the CLIP score (Hessel et al., 2021), LPIPS score (Zhang et al., 2018), and Aesthetic score (Schuhmann, 2022) to evaluate perceptual similarity and aesthetic quality, determining how well the generated textures align with real textures and input prompts. We additionally report inference time and giga floating-point operations (GFLOPs) to quantify computational efficiency, enabling fair comparison of runtime and compute cost across methods under fixed hardware, batch size, and resolution. The details for the metrics are provided in A.6. For our qualitative analysis, participants rated the results on a 1–5 scale according to two criteria defined in A.8: visual quality and text fidelity.

## 4.2 MAIN RESULTS

**Qualitative Comparisons** As shown in Figure 4, we evaluate all methods on five prompts targeting different stylization aspects: artistic pattern reproduction (Great Wave bottle), material rendering

(metal engraving cart), abstract style interpretation (fire-skin goldfish), translucent materials (crystal monkey), and complex multi-attribute handling (neon kettle). LVstyler consistently produces vibrant, detailed textures that faithfully capture the intended styles, while baselines show varying limitations such as pattern simplification, muted colors, or incomplete prompt interpretation.

Regarding baseline characteristics: Latent-NeRF tends to generate noisy textures with blurry surfaces; Text2Tex provides smoother outputs but with inconsistent textures at geometric edges; TEXTure produces clear inpainting but with occasional abrupt transitions on curved surfaces; Paint3D generates clean textures but sometimes exhibits multi-face artifacts. Among recent multi-view diffusion methods, SyncMVD achieves consistent texture application suited for photorealistic tasks but with uniform appearances; TEXGen offers UV-space generation but with varying quality; Hunyuan3D-2.0 provides stable material consistency with efficient inference, excelling in realistic texturing scenarios.

Table 1: Quantitative and qualitative comparisons on the text-to-texture task.

| Method | Perceptual Quality | | | | Efficiency | | User Study | |
|---|---|---|---|---|---|---|---|---|
| | FID $\downarrow$ | LPIPS $\downarrow$ | CLIP $\uparrow$ | Aesthetic $\uparrow$ | GFLOPs $\downarrow$ | Time (s) $\downarrow$ | Visual $\uparrow$ | Text Fidelity $\uparrow$ |
| Latent-NeRF (2023) | 236.48 | 0.1974 | 0.9009 | 4.1184 | $2.76 \times 10^5$ | 780.19 | 3.36 | 3.66 |
| Text2Tex (2023) | 206.08 | 0.1222 | 0.9073 | 4.3693 | $1.13 \times 10^6$ | 357.33 | 3.45 | 3.79 |
| TEXTure (2023) | 242.37 | 0.1645 | 0.9169 | 4.3055 | $3.02 \times 10^6$ | 71.56 | 3.72 | 4.02 |
| Paint3D (2023) | 195.87 | 0.1027 | 0.9080 | 4.3421 | $6.50 \times 10^2$ | 117.93 | 3.58 | 3.77 |
| SyncMVD (2024) | 187.33 | 0.1005 | 0.9140 | 4.3683 | $2.00 \times 10^3$ | 71.11 | 4.00 | 4.02 |
| TEXGen (2024) | 191.17 | 0.2485 | 0.9048 | 4.3675 | $2.53 \times 10^3$ | 61.55 | 3.53 | 3.77 |
| Hunyuan3D-2.0 (2025) | **176.32** | 0.1094 | 0.9047 | 4.3520 | $7.64 \times 10^3$ | **43.65** | 3.89 | 3.82 |
| LVstyler (Ours) | 191.05 | **0.1004** | **0.9161** | **4.3717** | $\mathbf{1.82 \times 10^3}$ | 65.21 | **4.14** | **4.05** |

**Quantitative Comparisons** In Table 1, we compare LVstyler against several state-of-the-art approaches in text-driven texture synthesis using key metrics. LVstyler achieves the lowest LPIPS (0.1004), indicating superior texture consistency. While achieving a high CLIP (0.9161), our method also excels with the highest Aesthetic (4.3717). These results demonstrate LVstyler's effectiveness in generating visually appealing textures aligned with the styling prompt compared to other methods. Notably, LVstyler also demonstrates superior efficiency with the lowest GFLOPs (16.25), significantly outperforming other methods in computational efficiency.

In contrast, **LVstyler** is specifically designed for artistic stylization, which explains our strengths in style fidelity and color vibrancy. The user study results (Table 1) confirm this: LVstyler achieves the highest scores in both Visual Quality (4.14) and Text Fidelity (4.05). Our method provides superior style fidelity through CLIP-guided optimization with dual LoRA architecture, robust text-to-texture alignment for complex prompts, and exceptional computational efficiency (16.25 GFLOPs) enabling rapid style exploration.

**User Study** We carried out a user study with 61 human responses to evaluate the randomly selected meshes and corresponding text prompts. Meanwhile, we leveraged visual language models to simulate 48 responses. As shown in Table 1, our method received the highest average scores in both criteria, demonstrating clear advantages in subjective visual quality and prompt consistency. More details on the user study can be found in Appendix A.8.

## 4.3 ABLATION STUDIES

### 4.3.1 IMAGE SPACE ONLINE LEARNING

We analyze the choice of epochs from two perspectives: the four loss curves and the visual performance of the stylized images. From Figure 5a, we observe that before epoch 200, the model is still at the early stage of learning the target style. At epochs 50 and 100, the target style is not yet clearly reflected in the outputs. Some features remain blurry, and the images exhibit excessive smearing with a lack of fine-grained details. From Figure 5b, we can see that the content loss $L_c$ converges at around 200 epochs, which means that the difference between original images and output images has stabilized. We conclude that the output images have incorporated the target style. After epoch

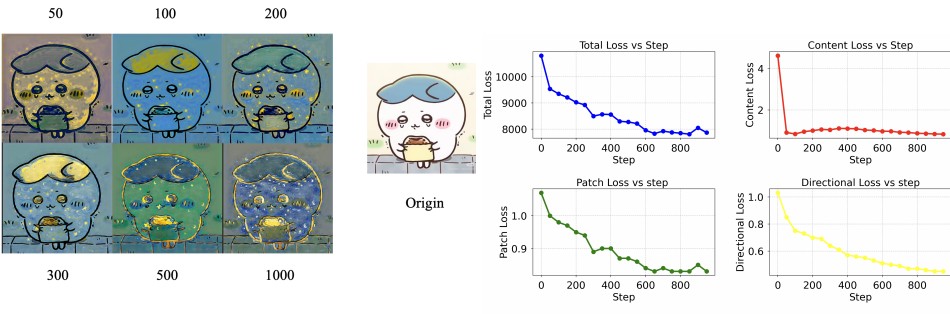

(a) Chiikawa styling results at different epochs. Prompts: Chiikawa in Van Gogh's style

(b) Comparison of losses at different steps

Figure 5: The iterated stylized graphs in (a) visually represent the output of the image styling network at different epochs. There are four subplots in (b): the total loss, content loss, patch loss, and directional loss, for loss comparisons across different epochs during image space online learning.

200, although the PatchCLIP loss $L_{\text{patch}}$ and directional CLIP loss $L_{\text{dir}}$ continue to decrease slightly, it does not imply that the model achieves a better trade-off between content preservation and style transfer. In fact, from Figure 5a, at epochs 300, 500, and 1000, the global tone and stylistic characteristics do not change much. For PatchCLIP, the overly small patch loss also leads to overfitting in some local regions. As shown in Figure 5a, at epochs 300, 500, and 1000, the eyes and hair of Chiikawa, as well as the bread she holds, are over-stylized. While we aim for the overall image to reflect the style of Starry Night, we do not intend for the core subject to have star-shaped eyes or galaxy-textured hair. In summary, epoch 200 is the optimal threshold for image space styling learning.

### 4.3.2 SHAPE LoRA

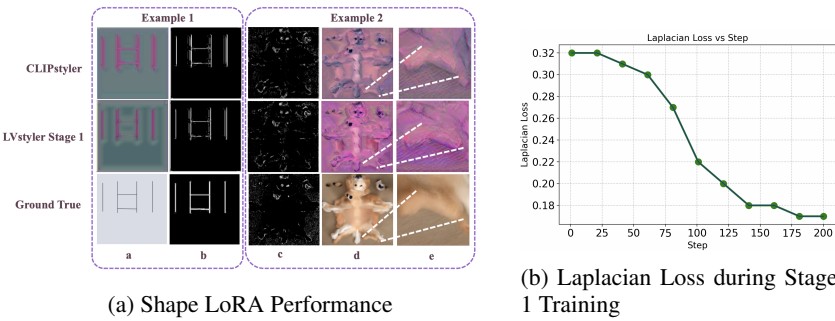

(a) Shape LoRA Performance

(b) Laplacian Loss during Stage 1 Training

Figure 6: For Stage 1: (a) Highlights the initial efficacy of the shape LoRA in preserving styling shapes. (b) Visualizes the Laplacian loss, illustrating loss optimization progress.

The shape LoRA is designed to produce clearer edge details for the UV texture styling. A fundamental limitation of CLIPstyler is its tendency to produce significant noise when applied directly to UV maps, as it lacks explicit structural constraints for preserving geometric boundaries. As shown by Example 1 in Figure 6a, compared to the CLIPstyler, LVstyler–Stage 1 delivers crisper, well-defined edges, while the zoom-ins of Example 2 in Figure 6a reveal richer local details and less noise. This visual improvement is supported by the Laplacian-edge loss in Figure 6b, which converges to a small value, indicating a low level of random noise, which is critical for obtaining high-quality textures for industrial usage.

### 4.3.3 UV LoRA

Integrating UV LoRA preserves the original UV map properties and details by restricting the style-learning process to the UV space. Since CLIPstyler operates exclusively in image space, it cannot handle UV-specific characteristics such as seam discontinuities, island boundaries, and non-uniform sampling, often resulting in over-stylization that destroys the independence of different UV regions.

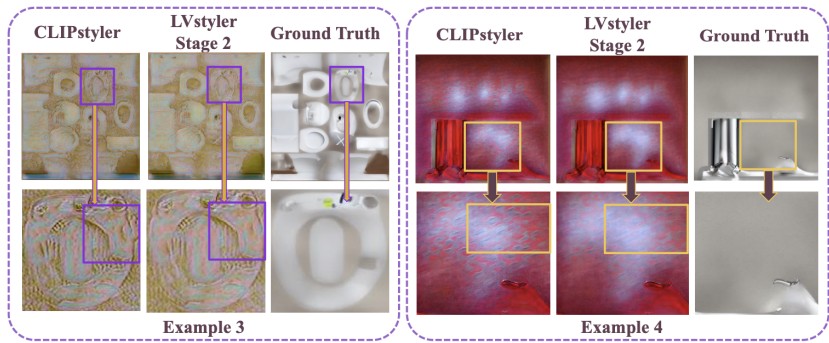

Figure 7: The UV LoRA performance at stage 2

Table 2: Comparisons of CLIPstyler, LVstyler (stage 1), and LVstyler (stage 2).

| Methods | Stage | PSNR ↑ | SSIM ↑ | LPIPS ↓ |
|---|---|---|---|---|
| CLIPstyler | – | 10.7917 | 0.38875 | 0.7682 |
| LVstyler | 1 | **11.6884** | 0.4440 | 0.7359 |
| LVstyler | 2 | 9.9598 | **0.4902** | **0.7359** |

As shown in Figure 7 Example 3, LVstyler maintains the surface smoothness between outputs and ground-truth images, in contrast to CLIPstyler's output with abrupt surfaces and artifacts. Moreover, the detailed zoom-ins in Figure 7 Example 4 demonstrate that LVstyler creates a more consistent UV texture with the original one, whereas the vanilla CLIPstyler produces spurious spots that disrupt the original structure.

Table 2 presents assessments of two stages of LVstyler alongside CLIPstyler by examining three key performance metrics: the peak signal-to-noise ratio (PSNR), structural similarity index (SSIM), and LPIPS. We compute PSNR using the original UV map as the reference, measuring pixel-level differences between generated and reference textures. LVstyler stage 1 achieves notable enhancement over CLIPstyler, particularly in terms of PSNR and SSIM, with scores of 11.6884 and 0.4440, respectively, indicating improved image fidelity and structural quality. While LVstyler stage 2 shows a slight decrease in PSNR to 9.9598, this is an intentional trade-off: Stage 2 actively amplifies style intensity through UV LoRA, causing the output to deliberately deviate from the original UV texture at the pixel level to achieve stronger style expression. Despite the PSNR decrease, Stage 2 compensates with the highest SSIM value of 0.4902, indicating superior preservation of structural details and better perceptual quality. Both stages of LVstyler maintain the same LPIPS score of 0.7359, demonstrating consistent perceptual similarity. Overall, this ablation underscores the explicit trade-off between style expression and pixel-wise reconstruction fidelity in LVstyler's iterative refinement process.

## 5 CONCLUSION

We propose LVstyler, a novel two-stage generative framework for flexible and efficient stylization of 3D UV maps. LVstyler incorporates the online learning styling model with two pretrained LoRA modules. It ensures shape consistency and adapts the model to the UV texture domain. This dual enhancement allows the model to handle more expressive prompts while preserving the structural integrity of the original object. Compared to state-of-the-art methods such as Latent-NeRF, TEXTure, Text2Tex, Paint3D, SyncMVD, TEXGen, and Hunyuan3D-2.0, our model achieves significant improvements across multiple metrics, including LPIPS, CLIP scores, and aesthetic quality, while maintaining superior computational efficiency. These results highlight the effectiveness of our approach in generating high-quality, style-consistent, and semantically aligned 3D textures.

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

## A  TECHNICAL APPENDICES AND SUPPLEMENTARY MATERIAL

### A.1  DESIGN PHILOSOPHY

Our method requires pre-existing UV texture maps, which reflect a deliberate design choice that aligns with the broader LoRA paradigm. Similar to how LoRA cannot generate images from scratch but enables precise fine-tuning of existing diffusion models, or how it facilitates concise responses from pre-trained LLMs, our LVstyler serves as a specialized refinement tool rather than a complete generation system. This design philosophy addresses a critical gap in the current 3D content creation pipeline. Numerous existing methods already produce 3D meshes with initial UV maps, including Paint3D, yet these often generate textures that lack the stylistic precision demanded by end users. Additionally, the industry possesses extensive libraries of human-designed meshes and UV assets that require re-styling for diverse applications. Our model provides the essential gateway that bridges the initialization phase of UV generation with the varied styling demands from end users in gaming, film, and design industries. Rather than viewing the UV map requirement as a limitation, it represents our focused approach to solving the "last mile" problem in 3D texture creation—transforming adequate but generic textures into precisely styled, high-quality assets that meet specific creative requirements. This specialization enables superior performance in the styling domain while maintaining compatibility with existing 3D content pipelines. While we do not explicitly disentangle geometry and lighting, LVstyler is deliberately scoped as a lightweight style-transfer complement to diffusion-based pipelines, prioritizing multi-view/seam robustness and structural preservation rather than relighting. In practice, our outputs can be combined with lighting-aware modules (e.g., Paint3D) to recover illumination realism, with LVstyler providing the final fine-grained stylistic refinement of 3D assets within standard content pipelines. Our code is placed on the Anonymous GitHub: `https://anonymous.4open.science/r/LVstyler-9952/README.md` and will be submitted as one of the supplementary materials.

### A.2  INPUT REQUIREMENTS AND ROBUSTNESS

A notable strength of LVstyler lies in its robustness to diverse input UV map sources, accommodating both human-defined and AI-generated textures without requiring pristine, artifact-free inputs. As illustrated in Figure A.1, our framework successfully processes two fundamentally different UV map categories. The top row demonstrates a *human-defined UV map*—a clean, manually crafted texture layout for a goldfish model exhibiting well-organized UV islands and minimal noise. The bottom row presents an *AI-generated UV map* from automated texture synthesis pipelines (e.g., Paint3D)—characterized by complex geometric patterns, seam discontinuities, inconsistent color distributions, and localized noise artifacts typical of multi-view diffusion-based generation. Despite these stark differences in input quality and topology, LVstyler consistently produces high-fidelity stylized outputs for both cases. For the human-defined input, our method transforms the simple orange texture into a vibrant fire-skin goldfish with rich color variation and flame-like patterns. For the AI-generated input with substantial artifacts, the framework robustly re-stylizes the noisy cart texture into a flowing water and light aesthetic, effectively suppressing input imperfections while introducing coherent artistic style. This robustness stems from our dual LoRA architecture: the shape LoRA module preserves essential geometric boundaries regardless of input noise levels, while the UV LoRA module learns domain-specific adaptations that respect the underlying UV coordinate system rather than amplifying existing artifacts. By training on diverse UV topologies encompassing both clean artist-designed maps and noisy AI-generated examples, our model develops invariance to spatial inconsistencies and artifact patterns. This characteristic substantially broadens the practical applicability of LVstyler, enabling seamless integration into automated content pipelines where input quality varies widely and cannot be guaranteed.

### A.3  ARCHITECTURE EFFECT ANALYSIS

**Resolution and Multiview Consistency Enhancement.** Our LoRA modules serve as constraint overseers that guide stylization direction while the core styling improvements, including resolution enhancement and multiview consistency, are achieved through our base model architecture.

For resolution enhancement, our base model employs asymmetric parameter control that enables output dimensions larger than input dimensions, with quality improvements accumulating across

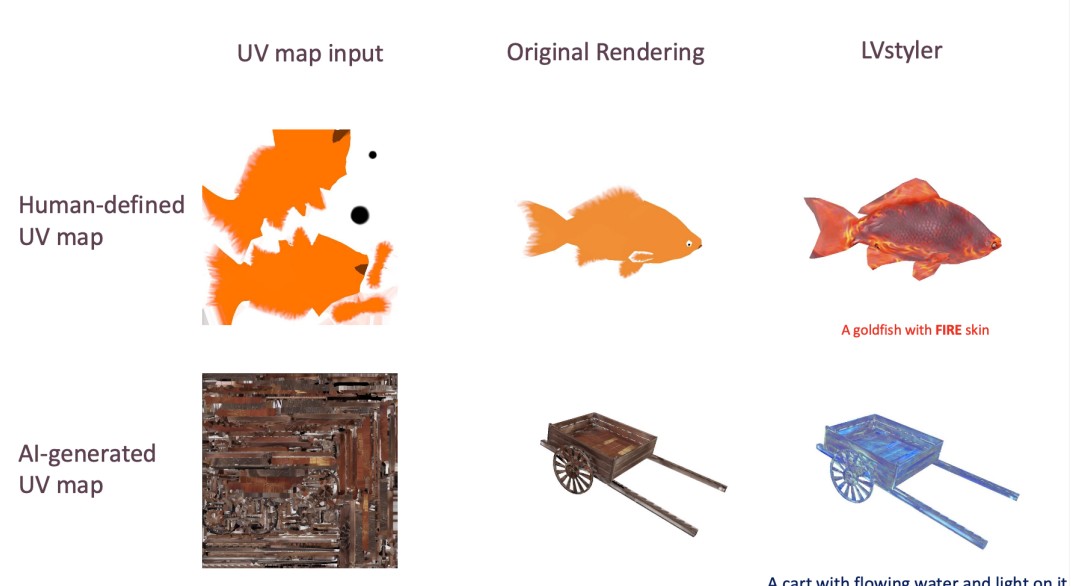

Figure A.1: Demonstration of LVstyler's robustness to diverse input UV map sources. Top row: Human-defined UV map (left: clean orange goldfish texture; center: original rendering; right: LVstyler result with fire skin style). Bottom row: AI-generated UV map with artifacts (left: noisy cart texture from automated pipeline; center: original rendering; right: LVstyler result with flowing water and light style). Despite significant differences in input quality and topology, LVstyler produces consistent, high-fidelity stylized outputs for both source types.

optimization iterations—higher resolutions require additional inference iterations for optimal convergence.

Regarding multiface consistency, unlike diffusion-based methods such as Paint3D that suffer from missing parts due to per-angle inpainting limitations of diffusion models, our approach directly optimizes over the entire 3D UV projection space, ensuring coherent styling across all viewpoints. According to the Figure A.2, this global optimization strategy eliminates the angular gaps and inconsistencies evident in Paint3D's bottle results, where certain viewing angles lack proper stylization due to the maximum capacity of inpainting diffusion methods, while our method maintains complete stylistic coverage through holistic UV space optimization.

## A.4 DATASETS

We utilize a subset of textured meshes from the Objaverse (Deitke et al., 2022) dataset, known for its large scale, diversity, and rich annotations in 3D modeling. Moreover, ModelNet (Wu et al., 2015) publishes the ModelNet40 which contains CAD point cloud models from 40 categories.

For the Objaverse dataset, we follow the method in Reich et al. (2023) to exclude meshes that lack textures and those depicting complex 3D scenes made up of multiple meshes. For ModelNet40, we randomly sample 3 objects from each class and convert the CAD model to a mesh representation with proper positioning augmentation. Following this preprocessing, the resulting dataset consists of approximately 410 objects from Objaverse, 120 objects from ModelNet40, and 140 synthetic textures from the texture inpainting frameworks. For each texture, masking is performed to extract the edges and noise details of UV maps, and image-styling models are applied to produce stylized images, constituting our LVstyler dataset.

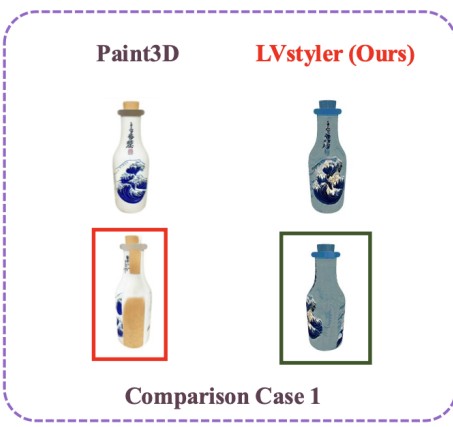 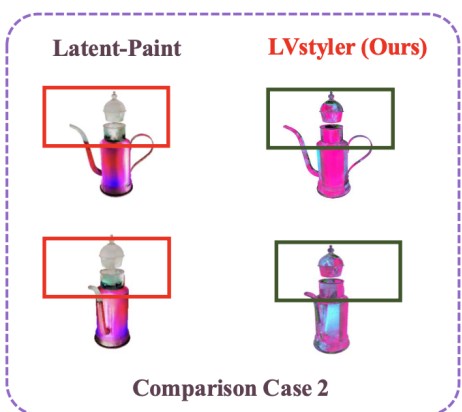

Figure A.2: Detailed comparison demonstrating how our LVstyler addresses multiview consistency challenges in complex 3D topology. The figure illustrates our method's capability to maintain coherent stylization across all viewpoints through global UV space optimization, contrasting with baseline approaches that suffer from angular gaps and inconsistent coverage due to per-angle inpainting limitations.

## A.5 IMPLEMENTATION

We train our LVstyler framework in a two-stage approach to effectively separate shape preservation and texture stylization capabilities. Each stage targets specific aspects of the UV texture stylization process with carefully calibrated loss functions.

### A.5.1 MODEL ARCHITECTURE

Our proposed model employs Low-Rank Adaptation (LoRA) modules attached to the backbone, with separate LoRAs for shape preservation (shape LoRA) and texture stylization (UV LoRA). The complete model consists of 615.27K trainable parameters, strategically designed to balance quality and efficiency. The model architecture follows a U-Net structure with cross-attention mechanisms for incorporating text guidance.

### A.5.2 STAGE 1: SHAPE LoRA TRAINING

In stage 1, we focus on training the shape LoRA module to preserve structural details while allowing for style adaptation. This stage emphasizes edge and shape integrity during the stylization process.

**Loss Functions**  Stage 1 employs a comprehensive loss function formulation,

$$\mathcal{L}_{\text{Stage1}} = \lambda_c \mathcal{L}_c + \lambda_{\text{tv}} \mathcal{L}_{\text{tv}} + \lambda_{\text{dir}} \mathcal{L}_{\text{dir}}$$
$$+ \lambda_{\text{patch}} \mathcal{L}_{\text{patch}} + \lambda_{\text{Laplacian}} \mathcal{L}_{\text{Laplacian}}.$$

The approach uses several key loss functions to guide image generation. The content loss $\mathcal{L}_c$ ensures feature similarity with a VGG-based model, weighted by $\lambda_c = 1.0$. The total variation loss $\mathcal{L}_{\text{tv}}$, weighted by $\lambda_{\text{tv}} = 2.0$, promotes spatial smoothness. The global CLIP directional loss $\mathcal{L}_{\text{dir}}$, weighted by $\lambda_{\text{patch}} = 9.0$, provides strong style guidance with $\lambda_{\text{dir}} = 500.0$, while the Patch-based CLIP directional loss $\mathcal{L}_{\text{patch}}$ ensures local style consistency. The Laplacian edge loss $\mathcal{L}_{\text{Laplacian}}$, crucial in stage 1, preserves shapes using a discrete $3 \times 3$ kernel $\begin{bmatrix} 0 & 1 & 0 \\ 1 & -4 & 1 \\ 0 & 1 & 0 \end{bmatrix}$, with $\lambda_{\text{Laplacian}} = 50.0$.

These losses work together for balanced content fidelity, smoothness, and structural integrity.

**Training Details**  We train with a batch size of 2 and an initial learning rate of $5e-4$, using a cosine decay schedule with warm-up steps. We process 64 random crops of size $128 \times 128$ for patch-based

Table 3: Comprehensive GPU Memory and Performance Summary.

| Description | Value |
|---|---|
| Allocated GPU Memory After Inference | 0.920 GB |
| Reserved GPU Memory After Inference | 3.619 GB |
| Max Allocated GPU Memory | 16.832 GB |
| Final Allocated GPU Memory | 0.933 GB |
| Model Loading Time | 3.29 seconds |
| LoRA Loading Time | 0.11 seconds |
| Inference Time | 60.58 seconds |
| Total Execution Time | 65.21 seconds |

losses per iteration. The training runs for 10 epochs, with checkpoints saved every 5 batches. The data are sourced from our curated dataset of UV textures with corresponding style prompts.

### A.5.3 STAGE 2: UV LoRA TRAINING

In stage 2, we train the UV LoRA module to focus on texture and appearance stylization, building upon the shape preservation capabilities established in stage 1.

**Loss Functions** Stage 2 modifies the loss function to emphasize style transfer,

$$\mathcal{L}_{\text{Stage2}} = \lambda_c \mathcal{L}_c + \lambda_{\text{tv}} \mathcal{L}_{\text{tv}} + \lambda_{\text{dir}} \mathcal{L}_{\text{dir}} + \lambda_{\text{patch}} \mathcal{L}_{\text{patch}} + \lambda_{\text{UV}} \mathcal{L}_{\text{UV}}.$$

Key changes to the configuration include a reduction in content preservation with $\lambda_c = 0.1$, down from the previous 1.0. Directional guidance is now lower, with $\lambda_{\text{dir}} = 50.0$ instead of 500.0. The patch-based loss has been reduced to $\lambda_{\text{patch}} = 5.0$, from 9.0. Additionally, a new term UV-specific regularization $\mathcal{L}_{\text{UV}}$ has been added, with a weight of $\lambda_{\text{UV}} = 0.01$. This configuration significantly reduces CLIP-based losses while increasing L1 image loss to better match the CLIPstyler-generated style target, allowing for more dramatic stylization while the shape LoRA maintains structural integrity.

**Training Details** Stage 2 uses the same batch size of 2 and a learning rate of $5e-4$. We initialize the model with the best checkpoint from stage 1, particularly the shape LoRA weights. Stage 2 training runs for 10 epochs, with more frequent checkpointing (every batch) to capture the rapid style adaptations.

### A.5.4 IMPLEMENTATION ENVIRONMENT

Our training infrastructure consists of either 16 RTX 4090 GPUs or 8 NVIDIA A800 GPUs, requiring approximately 10 hours to 10 hours to complete 200K steps (stage 1) and 5 hours for 100K steps (stage 2). We implement our framework using PyTorch. For inference, the model runs efficiently on a single RTX 4090 GPU and even many consumer-grade GPUs available in the market, making it practical for real-world applications.

**Shape LoRA Necessity Beyond Simple Masking** While masking might appear sufficient for edge preservation, it fundamentally fails for complex 3D objects due to several critical limitations. First, there exists no generalizable method to obtain perfect masks for complex 3D UV layouts with disconnected regions, overlapping elements, and varied topologies—manual creation is impractical while automated approaches consistently fail on intricate geometries. Second, binary masks inherently cause information loss by discarding necessary transitional regions that maintain visual coherence, effectively removing contextual details crucial for high-quality texture synthesis. Our shape LoRA addresses these limitations by learning soft structural constraints through continuous Laplacian-based optimization rather than hard binary boundaries. Unlike masks that provide static, context-unaware constraints, the shape LoRA adaptively preserves both sharp edges and gradual transitions while maintaining texture flow and structural relationships across the entire UV space. The ablation results demonstrate this advantage: the shape LoRA produces significantly crisper edges and richer local details with reduced noise compared to masking-based approaches, proving

that learned structural preservation outperforms binary constraint methods for complex 3D texture stylization tasks.

## A.6 EVALUATION METRICS

All metrics are computed on rendered images rather than UV maps. For each mesh we render two sets under identical cameras, lighting, backgrounds, resolution, and tone mapping. One set uses our synthesized texture and the other uses the original texture. FID is computed between these two render sets by extracting 2048-dimensional pool3 features from an Inception-V3 network and measuring the Fréchet distance over all renders aggregated across meshes and views. Lower is better. LPIPS is computed per view between a generated-texture render and the corresponding original-texture render using the official LPIPS implementation with linearized AlexNet weights, then averaged over views and meshes. Lower is better. CLIP score is computed per view as the cosine similarity between a frozen CLIP image encoder applied to the render and the CLIP text encoder applied to the styling prompt, then averaged over views and meshes. Higher is better. Aesthetic score is computed per render with the LAION aesthetics predictor applied to the image, then averaged over views and meshes. Higher is better. All evaluations use the same renderer, gamma, and color space conversions for both texture conditions to avoid domain shift, and we release the exact camera poses, environment settings, and scripts to reproduce these metrics. We report inference time and GFLOPs for a full end-to-end execution of one fine-grained texture generation run. Inference time is the wall-clock latency from the first model call to the final stylized UV output (i.e., all optimization or denoising passes included), excluding file I/O and rendering; we report the median over multiple runs after a short warm-up. Consistent with our focus on the re-stylization task, this measurement assumes that all 3D objects already possess initial UV textures—whether generated by existing methods or provided by artists—and captures only the time required to apply new styles, thereby reflecting the computational cost most relevant to industrial workflows where users frequently explore multiple style variants for a given object. GFLOPs are computed as the total floating-point operations over the entire execution at the evaluation resolution, summing all forward/backward passes (e.g., per-iteration updates for our method and full denoising schedules for diffusion baselines), and counting one multiply–add as two FLOPs.

## A.7 MORE COMPARISONS

**Baselines** We compare our method with state-of-the-art approaches, including Latent-NeRF (Li et al., 2023), TEXTure (Richardson et al., 2023), Text2Tex (Chen et al., 2023a), Paint3D (Zeng et al., 2023), SyncMVD (Liu et al., 2024b), TEXGen (Yu et al., 2024b), and Hunyuan3D-2.0 (Team, 2025). For methods without publicly available code (RomanTex (Feng et al., 2025)), we exclude them from our experimental comparison.

In the demo comparison we showed in the main paper, prompts are as follows: *decorative bottle with the Great Wave off Kanagawa by Hokusai, a trolley cart with metal engraving, a goldfish with fire skin, a green crystal Suzanne monkey, and a steaming kettle in a sleek, neon style featuring vivid pinks, electric blues, and radiant greens*. LVstyler demonstrates superior quality by producing varied texture maps and more details compared to baseline methods. We intentionally use these complex prompts consisting of object prompts and styling prompts, which helps to test the efficiency of different models handling the styling tasks.

In the experiments, we generate a 360-degree preview. We uniformly sample eight views to showcase all results side by side, enhancing the visualization of the output and demonstrating the model's robustness. The multiview results are shown in Figure A.3, A.4, A.5, and A.6.

## A.8 USER STUDY

In this study, we sought to assess the quality of generated textures through users' feedback. Participants were shown a series of images and asked to rate them using a Likert scale from 1 (very dissatisfied) to 5 (very satisfied) based on two criteria: **visual quality**, which describes how clear and natural the generated textures look, and **text fidelity**, which measures how well the textures match the input text prompts. The study included over 100 responses from 61 participants and 48 LLM agents. Each participant viewed a sequence of images depicting various texture styles from

our model and baseline models. The screenshots in Figure A.7 illustrate the questions posed to participants.

## A.9    MORE ABLATION RESULTS

The ablation results presented in Figures A.8 and A.9 demonstrate the effectiveness of our methods in enhancing texture quality. In Figure A.8, our approach significantly minimizes random noise across the UV texture. This reduction is crucial in producing high-quality textures with clearer edges and better overall clarity. Proceeding to Figure A.9, our final results after stage 2 reveal a substantial decrease in abnormal artifacts across different regions of the UV map. This change ensures the generation of superior textures, maintaining high quality and fidelity.

## A.10    LIMITATIONS AND BROADER IMPACTS

One limitation of our approach is its dependence on an existing UV texture map. However, this requirement constitutes a deliberate architectural decision rather than a fundamental constraint, as discussed in Section A.1. Our framework demonstrates remarkable capability in refining low-quality textures generated by upstream methods, functioning effectively on outputs from Paint3D (Zeng et al., 2023), TEXTure (Richardson et al., 2023), and Text2Tex (Reich et al., 2023), consistently achieving more diverse, detailed, and high-fidelity outcomes. The compact architecture of our model positions it as an efficient complement to existing research efforts, enabling fine-grained stylistic refinement while maintaining minimal computational overhead.

Another current limitation is the lack of fine-grained local control in styling, which remains a common challenge across existing methods. Local control is important for achieving more precise styling in specific parts of 3D subjects. To address this, a practical approach involves incorporating attention mechanisms to align 3D geometric data with the 2D projected UV texture. As part of our future work, we plan to integrate 3D geometric information into the framework to enable part-level disentanglement and more precise style manipulation in the 3D domain.

**Behavior Under Semantically Distant Prompts.** Our CLIP-guided optimization framework demonstrates robust performance across a wide spectrum of artistic styles, including semantically complex and culturally specific prompts. As evidenced in our qualitative comparisons (Figure 4), LVstyler successfully captures the distinctive characteristics of classical art movements—the swirling brushwork and vibrant color palette of Van Gogh's post-impressionist style, and the dynamic wave patterns with precise linework characteristic of Hokusai's ukiyo-e tradition. These results demonstrate that our method effectively handles prompts requiring nuanced interpretation of established artistic vocabularies. However, we observe degraded performance when prompts exceed approximately 20 simultaneous conditioning constraints (e.g., "a ceramic vase with Art Nouveau floral motifs, iridescent peacock feather patterns, gold leaf accents, crackle glaze texture, Japanese kintsugi repairs, Tiffany stained glass colors, and Mucha-style border decorations"). Such highly composite prompts present challenges for the underlying CLIP model, which struggles to maintain coherent alignment across numerous simultaneous stylistic objectives. In these cases, the optimization process may converge to solutions that emphasize certain style attributes while underrepresenting others, or produce outputs that average across the specified styles rather than harmoniously integrating them. This limitation reflects inherent constraints in current vision-language models rather than architectural deficiencies in our method, and future advances in compositional text-image understanding may directly benefit our framework.

Lastly, our method inherits the expressivity constraints of the underlying CLIP model, which may struggle with highly abstract or culturally-specific style concepts that fall outside its training distribution, potentially limiting the diversity of achievable stylistic expressions. Additionally, CLIP's text-image alignment capabilities can be inconsistent for complex multi-modal prompts that combine intricate object descriptions with nuanced artistic styles. To address these limitations, future work could integrate multiple vision-language models with complementary strengths, such as combining CLIP with specialized art-focused embeddings or incorporating fine-tuned domain-specific encoders that better capture artistic nuances and cultural styling elements.

Hobbyists and professionals alike can leverage our model to produce more stable, high-quality stylized images. While the framework is primarily designed as a professional-grade component for

image and 3D asset styling, with its clearest influence likely to appear first in academic research, it still offers meaningful broader impacts. Specifically, it streamlines the creation of consistent textures for indie game assets, VR/AR prototypes, rapid concept art, and educational visualizations, thereby making advanced stylization workflows accessible to a much wider creative community.

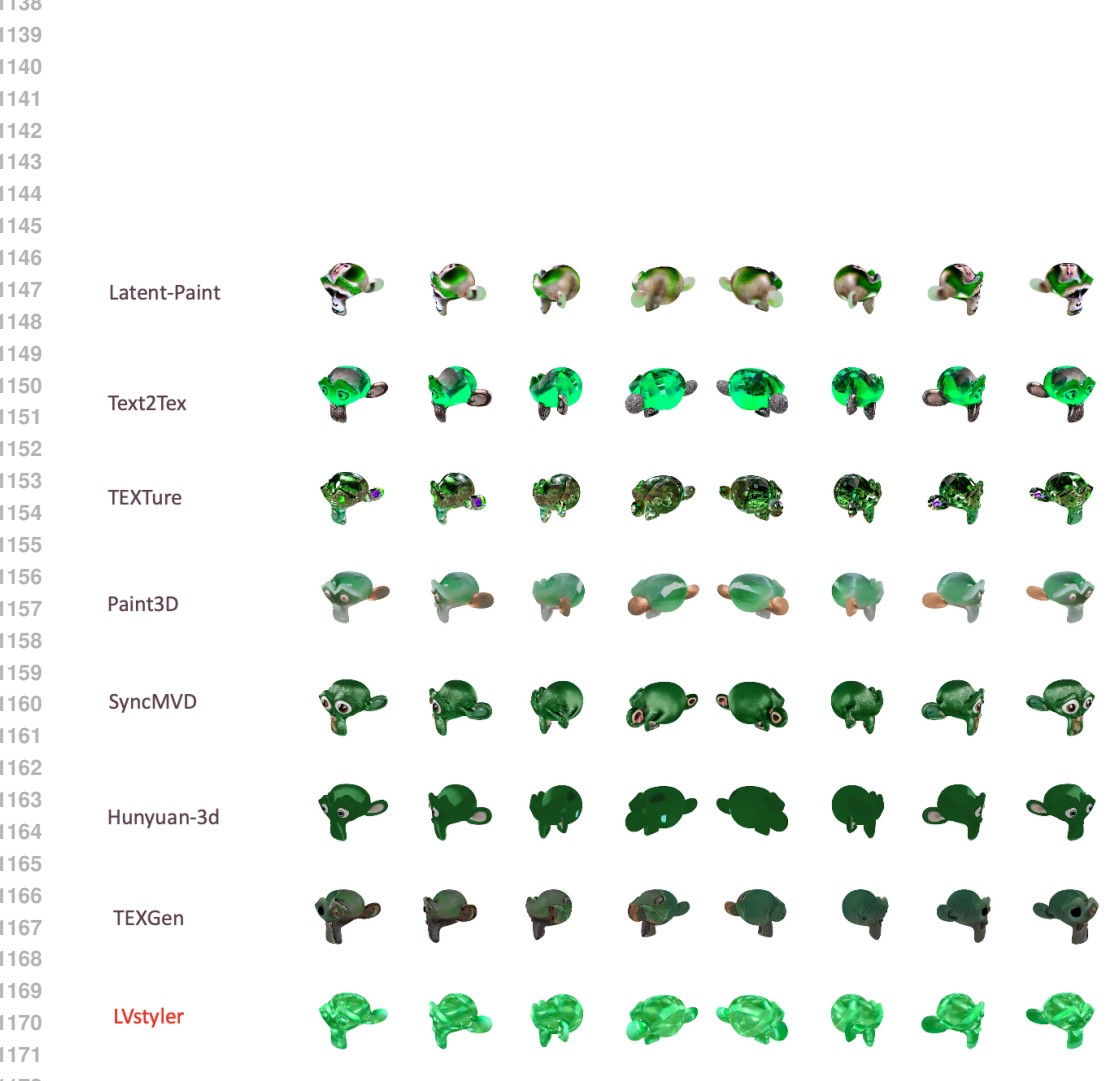

Latent-Paint

Text2Tex

TEXTure

Paint3D

SyncMVD

Hunyuan-3d

TEXGen

LVstyler

A **green crystal** Suzanne monkey

Figure A.3: Multiview comparison on the Suzanne monkey. Prompt: "a green crystal Suzanne monkey."

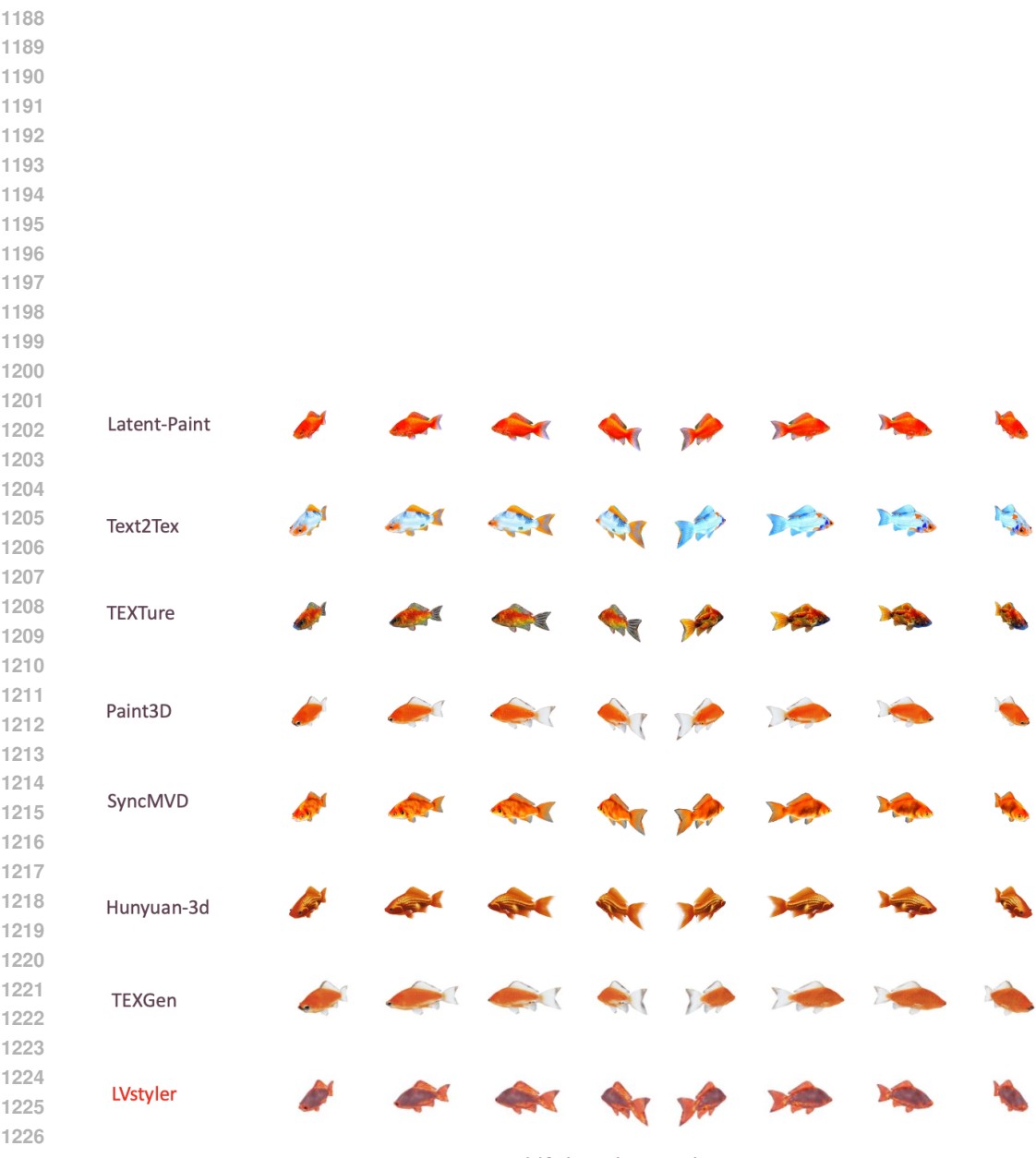

A goldfish with **FIRE** skin

Figure A.4: Multiview comparison on the goldfish. Prompt: "a goldfish with fire skin."

1242
1243
1244
1245
1246
1247
1248
1249
1250
1251
1252
1253
1254
1255
1256
1257
1258
1259
1260
1261
1262
1263
1264
1265
1266
1267
1268
1269
1270
1271
1272
1273
1274
1275
1276
1277
1278
1279
1280
1281
1282
1283
1284
1285
1286
1287
1288
1289
1290
1291
1292
1293
1294
1295

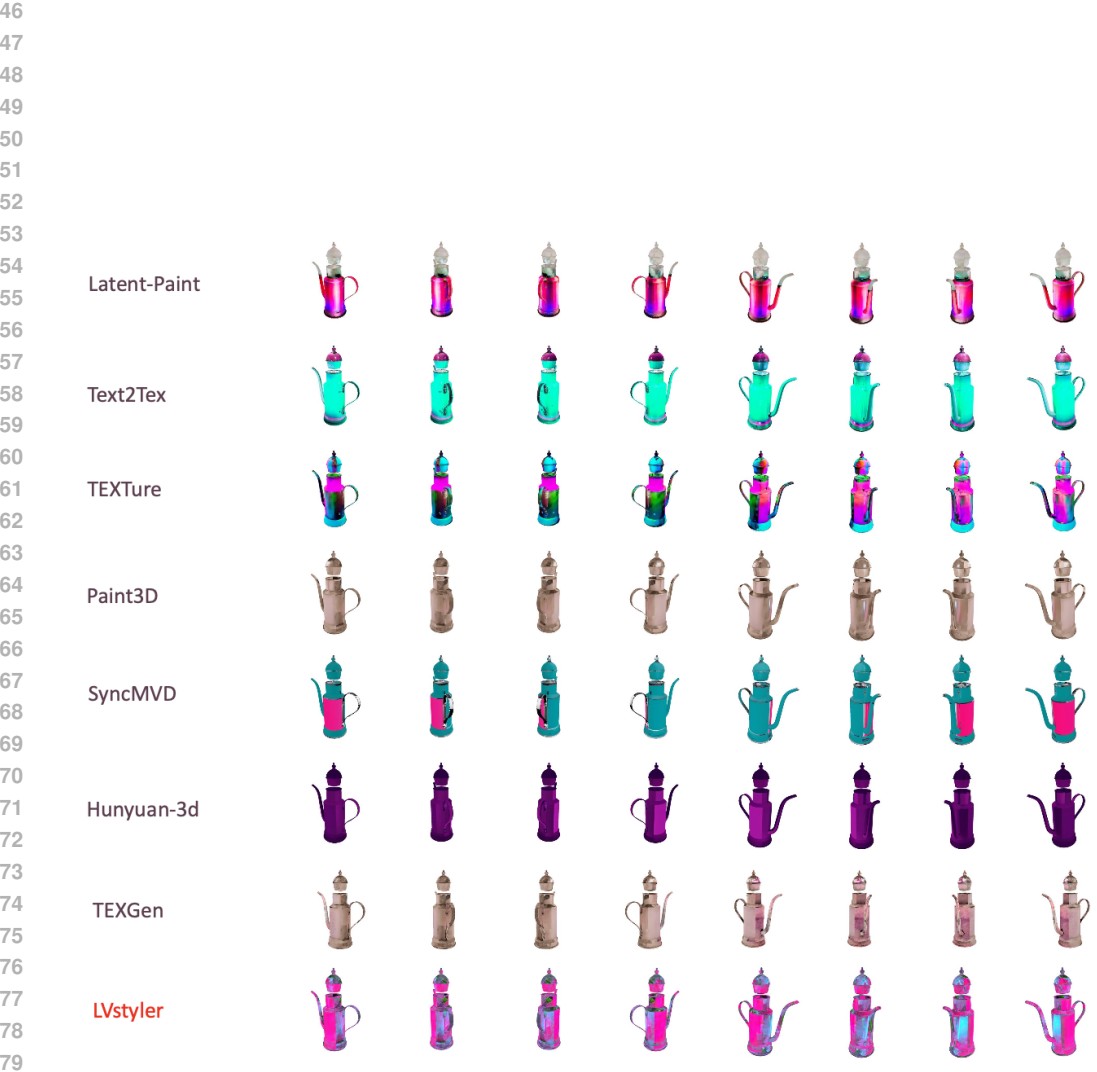

A steaming kettle in a **sleek, neon style**,
including **vivid pinks, electric blues, and radiant greens**

Figure A.5: Multiview comparison on the steaming kettle. Prompt: "a steaming kettle in a sleek neon style featuring vivid pinks, electric blues, and radiant greens."

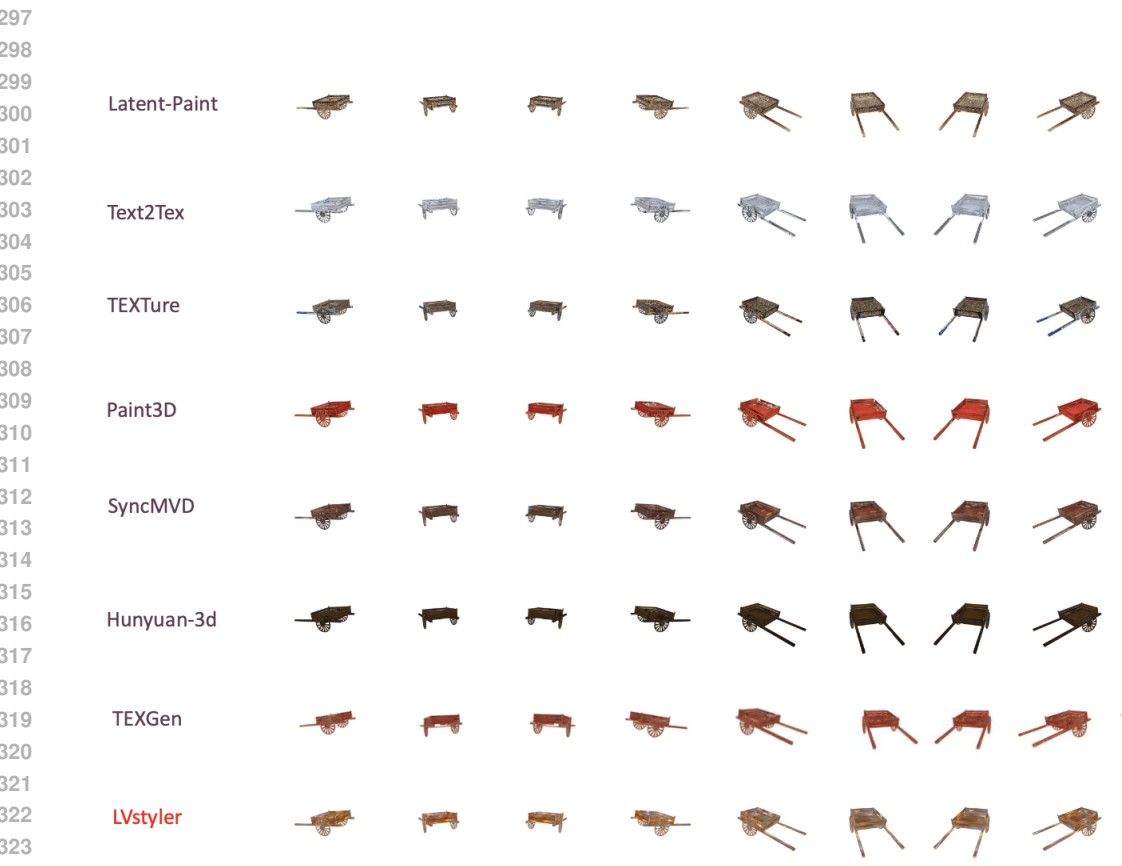

Figure A.6: Multiview comparison on the trolley cart. Prompt: "a trolley cart with metal engraving."

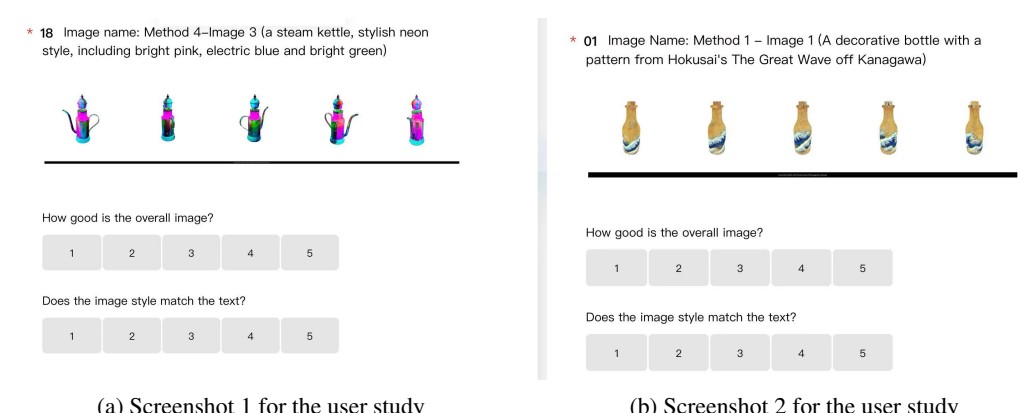

(a) Screenshot 1 for the user study      (b) Screenshot 2 for the user study

Figure A.7: The questions for the user study are i) How visually appealing is the image overall? ii) How well does the image style align with the description text?

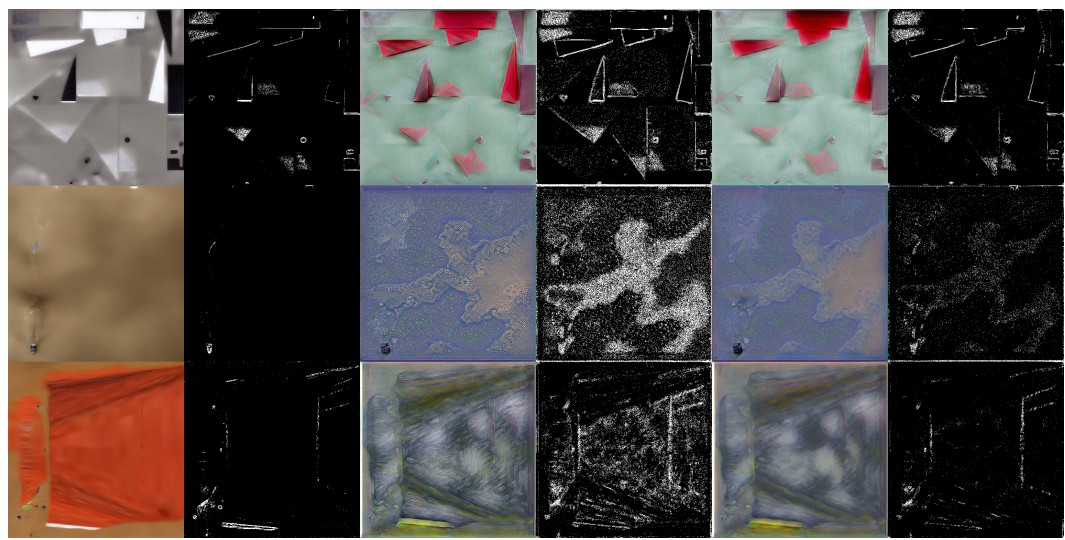

Figure A.8: Columns 1 and 2 are the original image and mask for the original texture, columns 3 and 4 are CLIPstyler outputs, and columns 5 and 6 are LVstyler outputs.

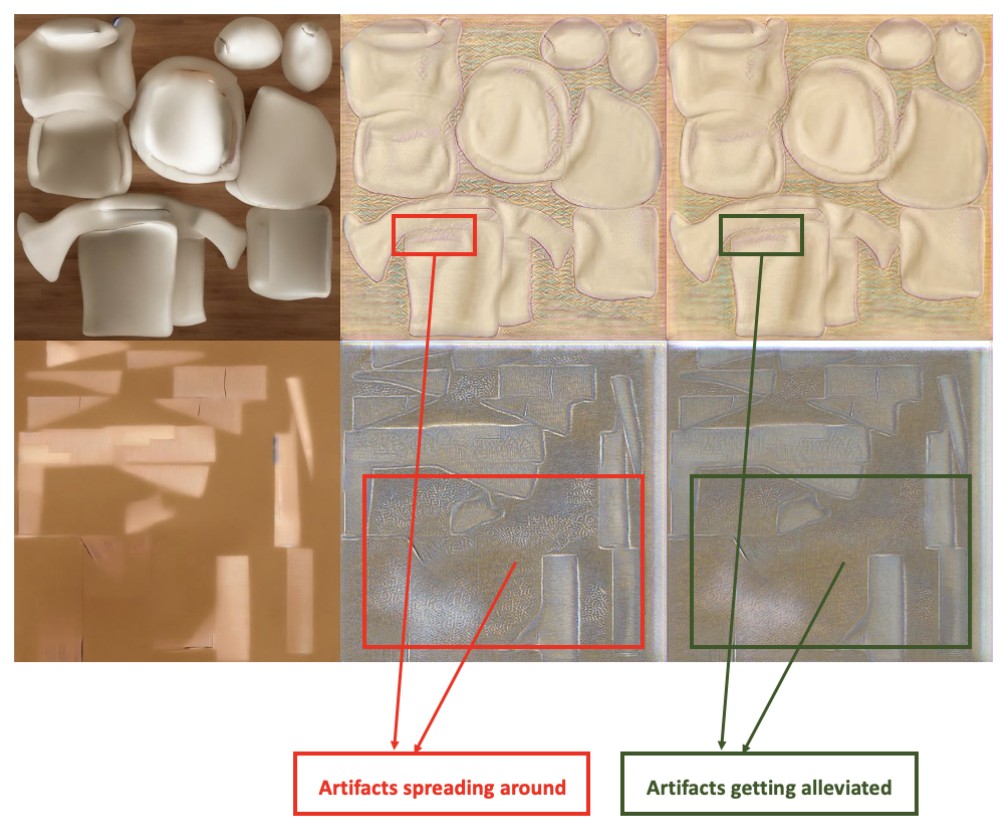

Figure A.9: The first column is the original textures, the second column is CLIPstyler outputs, and the third column is LVstyler outputs.

