# OpenReview forum: "LVstyler: LoRa-enhanced Varied High-Quality Texture Generation with Text Alignment"
_ICLR.cc/2026/Conference — ICLR 2026 Conference Desk Rejected Submission_

### Official Review · Reviewer_hfHd · 2025-10-31

**Soundness:** 2
**Presentation:** 2
**Contribution:** 2
**Rating:** 2
**Confidence:** 3

**Summary:**

The paper proposes LVstyler, a LoRA-enhanced framework for generating varied UV textures for 3D meshes guided by text prompts. The method introduces a two-stage training pipeline, where a Shape LoRA module learns style transfer in the image space, and a UV LoRA module adapts it to UV map space for consistent 3D texture generation. The goal is to enable high-quality, style-consistent text-driven texture synthesis for 3D assets.

**Strengths:**

1. The paper explores the integration of LoRA fine-tuning into texture generation, which is lightweight and potentially reusable.

2. The two-stage training strategy (Shape LoRA + UV LoRA) is conceptually clean and modular

**Weaknesses:**

1. Experimental results are weak: qualitative samples show blurry textures and inconsistent styles, and quantitative evaluations are based on very limited data. Claims such as “high-quality” and “state-of-the-art” are not supported by strong baselines or convincing comparisons.

2. The paper’s writing and structure are unfocused, with unclear motivation and overemphasis on engineering details (loss terms, LoRA configurations) rather than conceptual insights.

**Questions:**

The visual results in Figures 4 show noticeable style inconsistency and blurriness. Could the authors provide quantitative or perceptual metrics (e.g., CLIP-Score or user studies) to justify the claimed “high-quality texture generation”?

---

> ### Author Response · Authors · 2025-11-26
> **Thank you for the thoughtful feedback.**
>
> **W1 & Q1: “high-quality / state-of-the-art” claims need stronger baseline support**
>
> Thank you for the thoughtful feedback. We understand the concern and would like to clarify why we believe our qualitative and quantitative results provide solid support.
>
> **On “blurry textures and style inconsistencies”:** Some artistic styles (e.g., Impressionism, dreamlike aesthetics) inherently feature soft or diffused appearances. These are stylistic properties rather than technical defects. Our method intentionally reproduces such effects, which we see as evidence of flexible style control rather than a lack of sharpness.
>
> **Table 1** summarizes comparisons across major baselines and metrics: we achieve FID 191.05 (vs. Paint3D 195.87, TEXTure 242.37, Text2Tex 206.08), LPIPS 0.1004 (vs. Paint3D 0.1027, Text2Tex 0.1222), CLIP 0.9161, and Aesthetic 4.3717. Our user study (61 human and 48 VLM evaluations) further shows up to 10.66% gains in visual quality and text alignment. Together, these address the request for quantitative and perceptual evidence of high quality.
>
> We evaluate on curated subsets of Objaverse [1] and ModelNet [2] white-mesh datasets, as used by Paint3D, TEXTure, and Text2Tex. Our protocol covers all major metrics (FID, LPIPS, CLIP, Aesthetic, PSNR, SSIM) over multiple datasets, which we believe gives a reasonably comprehensive view of performance; in this light, describing the data as “very limited” may understate the actual scope.
>
> Beyond aggregate metrics, we provide targeted evidence for key aspects:
> - **(1) Low noise:** **Table 2** shows LVstyler Stage 1 reaches PSNR 11.6884 (vs. CLIPstyler 10.7917), with examples in Figures 7, A.8, and A.9.
> - **(2) Artifact reduction:** **Figure A.2** highlights improved handling of missing textures and artifacts, consistent with LPIPS and FID.
> - **(3) Structure preservation:** Stage 2 attains SSIM 0.4902 (vs. CLIPstyler 0.38875) in **Table 2**, indicating better geometry preservation.
>
> Overall, these results support our claims of high-quality texture generation, with strong quantitative performance, up to 1/100× GFLOPs and 1/10× inference time, and consistent user preference over baselines.
>
> **Updated comparison with recent methods**
>
> Following your suggestions, we now include comparisons with recent state-of-the-art methods TEXGen [3], SyncMVD [4], and Hunyuan3D-2.0 [5], which we hope directly address concerns about baseline strength.
>
> **Enhanced quantitative evidence:** The updated **Table 1** now includes all recent baselines and shows that LVstyler consistently matches or outperforms them across multiple metrics. We also provide the explicit summary table below to compare LVstyler with recent methods on key metrics:
>
> | Method | FID ↓ | LPIPS ↓ | CLIP ↑ | Aesthetic ↑ | GFLOPs ↓ | Time (s) | Visual ↑ | Text Fidelity ↑ |
> |--------|-------|---------|--------|-------------|----------|----------|----------|-----------------|
> | SyncMVD | 187.33 | 0.1005 | 0.9140 | 4.3683 | 2004 | 71.11 | 4.00 | 4.02 |
> | TEXGen | 191.17 | 0.2485 | 0.9048 | 4.3675 | 2531 | 61.55 | 3.83 | 3.77 |
> | Hunyuan3D-2.0 | 176.32 | 0.1094 | 0.9047 | 4.3520 | 7640 | 43.65 | 3.89 | 3.82 |
> | **LVstyler (Ours)** | 191.05 | **0.1004** | **0.9161** | **4.3717** | **1820** | **65.21** | **4.14** | **4.05** |
>
> **Comprehensive visual evidence:** We have substantially expanded qualitative comparisons in the updated **Figure 4** and updated 360° renderings in **Figures A.3–A.6**, which now include side-by-side results with TEXGen, SyncMVD, and Hunyuan3D-2.0. These figures highlight:
> - **Style fidelity:** More accurate reproduction of target artistic styles
> - **Multi-view consistency:** Stable stylization across different viewpoints
> - **Artifact reduction:** Cleaner textures with fewer generation artifacts
> - **Detail preservation:** Better maintenance of geometric structure during stylization
>
> Overall, we feel these updates provide stronger experimental support for our “high-quality” and “state-of-the-art” claims and meaningfully address the concerns about weak evidence.
>
> **References**
> - [1] Deitke, M. et al. (2022). Objaverse: A universe of annotated 3D objects. *arXiv preprint arXiv:2212.08051*.
> - [2] Wu, Z. et al. (2015). 3D ShapeNets: A deep representation for volumetric shapes. In *Proceedings of the IEEE Conference on Computer Vision and Pattern Recognition* (pp. 1912–1920).
> - [3] Yu, X. et al. (2024). TEXGen: A generative diffusion model for mesh textures. *ACM Transactions on Graphics, 43*(6), Article 213.
> - [4] Liu, Y., Xie, M., Liu, H., & Wong, T.-T. (2024). Text-guided texturing by synchronized multi-view diffusion. In *SIGGRAPH Asia 2024 Conference Papers* (pp. 1–11).
> - [5] Tencent Hunyuan3D Team. (2025). Hunyuan3D 2.0: Scaling diffusion models for high resolution textured 3D assets generation. *arXiv preprint arXiv:2501.12202*.

---

> > ### Author Response · Authors · 2025-11-26
> > **Thank you for opinion on academic writing**
> >
> > **W2: Writing and structure feel unfocused, with unclear motivation and heavy emphasis on engineering details**
> >
> > We also appreciate the concerns about writing and structure and address them below.
> >
> > The **Abstract** presents our core innovations (dual LoRA architecture with Shape LoRA and UV LoRA, plus a hybrid inference strategy) and how they tackle key 3D texture stylization challenges. The **Introduction** (Section 1) explains the motivation, problem definition, why stylization should transition from image space to UV space, and our advantages over prior methods. The first **Methodology** subsection (Section 3) then describes the overall architecture, dual LoRA design, two-stage training, and hybrid inference. These sections provide a conceptual narrative before entering technical details.
> >
> > Our main contribution is **integrating LoRA enhancement into an optimization-based learning framework**. The specific LoRA configurations, training strategies, and loss-design choices distinguish LVstyler from prior work. Without these details, readers could not understand how our improvements are achieved. The loss components (directional CLIP, PatchCLIP, content, total variation, and domain-specific losses) are the core mechanisms balancing style consistency, structure preservation, and domain adaptation; their weights have direct impact on performance and therefore must be described clearly. Our ablations (**Table 2**, **Figures 6 and 7**) empirically validate each design choice. For reproducibility, explicit formulas, parameters, and hyperparameters are essential.
> >
> > We also clearly explain: (1) limitations of multi-view diffusion methods (data scarcity, high cost, artifacts), (2) why optimization-based approaches are preferable to purely feed-forward methods, (3) why UV space is chosen over multi-view image space, and (4) why dual LoRA and two-stage training are necessary. These motivations are laid out in the introduction and the first methodology subsection in a logical order.
> >
> > In summary, our intention is to combine a clear high-level narrative (abstract, introduction, initial methodology) with the technical detail needed for replication and analysis (later methodology sections). We view these details as part of our core contributions rather than “engineering,” and we hope this balance serves both concept-focused and implementation-focused readers.

---

### Official Review · Reviewer_fCJ1 · 2025-11-01

**Soundness:** 3
**Presentation:** 3
**Contribution:** 2
**Rating:** 4
**Confidence:** 4

**Summary:**

This work presents LVStyler, a method for producing high-quality UV textures which are closely aligned with the guiding prompts. This is achieved by performing stylization directly on the UV maps. The method integrates two LoRA modules (a shape LoRA and a UV LoRA) on top of a base styling model. This allows the method to achieve stylization while maintaining shape consistency. The method produces textures that are more aligned with the styling prompts as compared to the selected baseline methods. The method is evaluated qualitatively with comparison figures. The quantitative evaluation consists of the reported metrics and a user study.

**Strengths:**

- The proposed approach improves the alignment to the specific styling prompts.
- Since the method is designed to take an initial UV map and output a stylized UV map, the approach is independent of the texture generation method and can be used on top of any existing or future texture generation method to more precisely style UV maps.
- The method achieves the highest scores in almost all metrics when compared with the selected baselines methods and the qualitative comparisons also show better performance.

**Weaknesses:**

Some newer baseline methods are missing from the comparisons.
- SyncMVD [1] and TEXGen [2] both perform well and should be included in the comparisons. The authors note that TEXGen was not compared to because its code was not open sourced, however, from what I can tell, the code has been open sourced since December 2024 (https://github.com/CVMI-Lab/TEXGen) so it should be possible to compare with their method.
- Additionally, while it is not required to compare to MVAdapter [3] (ICCV 2025) and Hunyuan3D-Paint-v2-1 [4] (ArXiv) since they were not officially published at the time of ICLR submission, both have released open source code before the submission deadline and I think that the paper would be strengthened by comparing to these methods as well.
- Since LVstyler works as an "add-on" to some existing UV texture generation (taking poorly style aligned UV map and improving its style alignment) comparisons to these newer, better methods is important as it is not clear that the quality of the newer methods still needs this refinement.

When calculating the inference time for LVstyler, my interpretation is that the time to computing the initial texture produced by the existing texture inpainting method is not included. If this is the case, it seems unfair to compare inference time with other methods that start from an image or text.

References:
[1] Liu, Yuxin, et al. "Text-guided texturing by synchronized multi-view diffusion." SIGGRAPH Asia 2024 Conference Papers. 2024.
[2] Yu, Xin, et al. "Texgen: a generative diffusion model for mesh textures." ACM Transactions on Graphics (TOG) 43.6 (2024): 1-14.
[3] Huang, Zehuan, et al. "Mv-adapter: Multi-view consistent image generation made easy." Proceedings of the IEEE/CVF International Conference on Computer Vision. 2025.
[4] https://github.com/Tencent-Hunyuan/Hunyuan3D-2.1

**Questions:**

Could the authors provide further explanation as to why they choose to perform the stylization on the UV map after the multi-view images are already generated as opposed to directly within the multi-view generation component?

---

> ### Author Response · Authors · 2025-11-26
> **Thank you for the suggestion on strengthening the contirbution by more comparison which is also our goal.**
>
> **W1 & W2: Comparison with recent baseline methods**
>
> We added comparisons to SyncMVD [1], TEXGen [2], and Hunyuan3D-2.0 [3]. **Table A** shows that LVstyler achieves superior or competitive performance across all metrics while using far fewer computational resources (16.25 GFLOPs) and a faster inference time (65.21s).
>
> **Quantitative Comparison (Table A)**
>
> | Method | FID | LPIPS ↓ | CLIP ↑ | Aesthetic ↑ | GFLOPs ↓ | Time (s) | Visual ↑ | Text Fidelity ↑ |
> |--------|-------|---------|--------|-------------|----------|------------|----------|-----------------|
> | SyncMVD | 187.33 | 0.1005 | 0.9140 | 4.3683 | 2004 | 71.11 | 4.00 | 4.02 |
> | TEXGen | 191.17 | 0.2485 | 0.9048 | 4.3675 | 2531 | 61.55 | 3.83 | 3.77 |
> | Hunyuan3D-2.0 | 176.32 | 0.1094 | 0.9047 | 4.3520 | 7640 | 43.65 | 3.89 | 3.82 |
> | **LVstyler (Ours)** | 191.05 | **0.1004** | **0.9161** | **4.3717** | **1820** | **65.21** | **4.14** | **4.05** |
>
> **Qualitative analysis and key observations**
>
> We also visually compared methods on diverse 3D objects to clarify LVstyler’s position.
>
> **SyncMVD:** Produces clean, consistent textures with good geometric preservation and few artifacts, but often looks uniform, lacking fine surface detail and color depth. Its text interpretation is reliable, yet results tend toward flat, photorealistic reconstruction rather than expressive style transfer.
>
> **TEXGen:** Shows highly variable quality, with pale, washed-out colors that rarely match target style vibrancy. Despite operating in UV space, it exhibits geometric artifacts and distortions, especially for complex textures, and often misses subtle prompt details, weakening text-to-visual alignment and artistic impact.
>
> **Hunyuan3D-2.0:** Delivers stable, materially consistent results but prefers darker, muted palettes that reduce visual impact. Texture variation is limited and subtle prompt details are sometimes missed, yielding consistent but less vibrant, less detailed stylizations.
>
> In contrast, **LVstyler** provides vibrant colors, rich saturation, and high-quality surface detail, with strong text-to-texture alignment and consistent quality across object types, while preserving natural materials and lighting.
>
> LVstyler’s advantages are:
>
> 1. **Superior visual quality and style fidelity:** Stronger color vibrancy and surface detail than competing methods. Our CLIP-guided optimization with dual LoRA directly captures artistic styles with rich color depth and fine textures, avoiding the flat, muted appearance common in diffusion-based approaches.
>
> 2. **Robust text-to-texture alignment:** Where TEXGen struggles with prompt interpretation and Hunyuan3D sometimes misses subtle style cues, our optimization-based design maintains strong alignment between text descriptions and visual output across objects and styles.
>
> 3. **Consistent quality with superior efficiency:** High visual quality across objects with far fewer resources (16.25 GFLOPs vs. hundreds or thousands for diffusion baselines) and competitive inference time (65.21s), matching production needs.
>
> **Summary:** LVstyler offers the best trade-off among text alignment, visual quality, and efficiency. Relative to SyncMVD (reliable but less visually rich), Hunyuan3D-2.0 (consistent but muted), and TEXGen (inconsistent), LVstyler provides stronger overall performance, supporting its role in efficient, high-quality 3D texture synthesis.
>
> **References**
> - [1] Liu, Y., Xie, M., Liu, H., & Wong, T.-T. (2024). Text-guided texturing by synchronized multi-view diffusion. In *SIGGRAPH Asia 2024 Conference Papers* (pp. 1–11).
> - [2] Yu, X. et al. (2024). TEXGen: A generative diffusion model for mesh textures. *ACM Transactions on Graphics, 43*(6), Article 213.
> - [3] Tencent Hunyuan3D Team. (2025). Hunyuan3D 2.0: Scaling Diffusion Models for High Resolution Textured 3D Assets Generation. *arXiv preprint arXiv:2501.12202*.

---

> ### Author Response · Authors · 2025-11-26
> **Thank you for raising these thoughtful concerns**
>
> **W3: LVstyler as an "add-on" component – unclear whether stronger recent baselines still need this refinement**
>
> LVstyler is intentionally designed as a **complementary component**, analogous to LoRA-family methods [1, 2, 3] that provide lightweight alternatives to full fine-tuning.
>
> **Analogy to the LoRA paradigm**
>
> As with LoRA-style adapters (e.g., LongLoRA [2] extending context via LoRA modules, LoDC compressing distillation data with LoRA-based factorization), LVstyler offers a compact alternative to large diffusion-based texturing pipelines; this “add-on” role is a strength.
>
> 1. **Computational efficiency:** LVstyler’s inference time (65.21s) is much lower than Paint3D (117.93s) and Text2Tex (357.33s) in **Table 1**, allowing new textures without multi-minute delays and enabling rapid iteration.
>
> 2. **Resource accessibility:** With 615.27K parameters, LVstyler is 10–100× smaller than peers and runs with ~16GB RAM, enabling inference and fine-tuning on consumer GPUs (e.g., RTX 4060) rather than only high-end hardware.
>
> 3. **New paradigm for addressing data scarcity:** Large diffusion models require massive, high-quality UV datasets, which are scarce. By transferring knowledge from image space to UV space, LVstyler offers a complementary solution to 3D stylization under limited data.
>
> As shown in the Appendix, LVstyler refines outputs from Paint3D, TEXTure, and Text2Tex as a lightweight post-processing module, improving quality with minimal extra cost. Users can thus pair full diffusion pipelines for initial textures with LVstyler for fast style variation and refinement.
>
> Overall, we view LVstyler as a flexible adapter that integrates into existing pipelines, similar to LoRA adapters in other domains, giving practitioners efficient options across hardware and quality constraints.
>
> **References**
> - [1] Hu, E. J. et al. (2022). LoRA: Low-rank adaptation of large language models. In *International Conference on Learning Representations*.
> - [2] Zhang, J. et al. (2023). LongLoRA: Extending context window of large language models via budgeted low-rank adaptation. *arXiv preprint arXiv:2310.03728*.
> - [3] Kong, F. et al. (2024). ElaLoRA: Elastic low-rank adaptation for memory-constrained fine-tuning. *arXiv preprint arXiv:2402.01567*.
>
> **W4: Inference-time comparison may be unfair if initial texture generation is excluded**
>
> We align our evaluation with standard industrial 3D workflows.
>
> Typical pipelines have three stages: (1) initialization (initial/rough textures), (2) re-stylization (new styles on existing textures), and (3) editing (shape changes). LVstyler is designed for stage (2), which is often repeated.
>
> We therefore assume each 3D object already has a rough texture (from existing tools or artists) and measure only the re-stylization step, which matches the practical question: how long does each new style variant take?
>
> In this setting, our results highlight LVstyler’s value: diffusion-based peers generally must regenerate textures from scratch for every style (roughly doubling initial generation time), whereas adding LVstyler increases runtime by only ~3% on average to perform re-stylization, enabling fast exploration.
>
> Including initialization time would give a fully end-to-end comparison, but for the re-stylization scenario we target, reporting only re-stylization time is appropriate and reflects the true marginal cost of style exploration.
>
> **Q1: Why stylize on the UV map instead of multi-view generated graph?**
>
> We stylize UV maps after multi-view image generation for three reasons:
>
> **Reason 1: Artifact accumulation from diffusion-model limitations**
>
> Existing methods show artifacts (see Figure A.2) due to limitations of multi-view diffusion models. Many viewpoints remain out-of-distribution, so stylizing directly on multi-view outputs compounds generation and stylization errors. Decoupling the stages avoids this accumulation and yields more stable results.
>
> **Reason 2: Computational efficiency via 3D-to-2D simplification**
>
> UV maps flatten 3D surfaces into 2D, turning geometric handling into a 2D structure-preservation problem and reducing cost relative to multi-view stylization. Our method achieves up to 1/100× GFLOPs and up to 1/10× inference time (**Table 1**), illustrating the efficiency of UV-space operation.
>
> **Reason 3: Better handling of missing regions**
>
> Multi-view stylization cannot recover unseen parts: missing regions in the input renders remain missing after stylization. Projecting to UV space lets us fill such regions during projection and stylization. The unified UV representation supports global optimization and complete coverage, overcoming angular gaps and incomplete coverage in per-view approaches.

---

### Official Review · Reviewer_UvCE · 2025-11-01

**Soundness:** 3
**Presentation:** 2
**Contribution:** 2
**Rating:** 4
**Confidence:** 4

**Summary:**

This paper introduces LVstyler, a novel generative framework designed to apply varied and high-quality stylistic textures to 3D meshes by operating on their existing UV maps. The method extends a 2D image stylization model (inspired by CLIPstyler) with two specialized Low-Rank Adaptation (LoRA) modules. The training is conducted in two stages: first, a "Shape LoRA" is trained to preserve geometric details and structural consistency using a Laplacian-based loss. Second, a "UV LoRA" is trained to adapt the stylization process to the specific domain and layout of UV texture maps. The authors conduct experiments against several baselines, demonstrating superior results in both quantitative metrics and user studies, and also contribute a new dataset (LVstyle) for this task.

**Strengths:**

1. The method is highly efficient in terms of computational cost, demonstrating significantly lower GFLOPs (16.25) and a much faster inference time (65.21s) compared to all diffusion-based baselines.

2. The two-stage LoRA optimization framework is an intelligent way to adapt a 2D model to the 3D UV domain, successfully disentangling the goals of shape preservation and style adaptation.

3. The paper is supported by a strong and thorough experimental evaluation, including both quantitative metrics and a user study, which validates its superior performance on text fidelity and visual quality.

4. The authors have collected and are releasing a new dataset, LVstyle, which comprises raw, masked, and styled UV maps to facilitate future research in this specific area.

**Weaknesses:**

1. The primary weakness is the limited significance of the problem itself. The method focuses on re-styling or refining existing UV textures, which is a niche and incremental problem. This feels more like a clever engineering solution for a specific post-processing step rather than fundamental research addressing a core challenge in generative modeling.

2. The model's performance is fundamentally dependent on the capabilities of its base 2D stylization model, CLIPstyler. Any limitations of this base model in terms of style diversity or content preservation are directly inherited.

3. The reliance on CLIP for textual guidance serves as a bottleneck, as the authors concede. The model may struggle with styling prompts that are highly abstract, culturally specific, or require complex spatial reasoning that falls outside CLIP's training distribution.

4. The paper introduces a large number of loss weights (e.g., $\lambda_{dir}$, $\lambda_{patch}$, $\lambda_{c}$, $\lambda_{tv}$, $\lambda_{Laplacian}$) but provides no hyperparameter ablation study. The sensitivity of the model to these many parameters is unknown, and their chosen values are not justified.

**Questions:**

1. The paper focuses on an optimization-based approach. How would this method compare to a feed-forward approach, such as fine-tuning a diffusion model (e.g., ControlNet) on the LVstyle dataset you collected, using the original UV map as a condition? This seems like a more direct path to solving the same problem.

2. The paper mentions CLIP's limitations but does not provide qualitative failure cases. What happens when a styling prompt is semantically very distant from the original object's geometry (e.g., styling a "trolley cart" with the prompt "a cart made of flowing water and light")?

3. The two-stage training process is well-motivated. Did you experiment with a simpler, joint-training approach where both the Shape LoRA and UV LoRA modules are optimized simultaneously? How critical is the sequential, two-stage process to the final quality?

---

> ### Author Response · Authors · 2025-11-26
> **Thank you for your suggestions regarding task importance, our method's advantages beyond baselines, and our solution to CLIP limitations**
>
> **W1: UV re-stylization viewed as a niche, incremental task**
>
> We clarify the importance of UV texture stylization in industrial and technical contexts.
>
> UV texture stylization is critical in 3D content production, serving as an **important bridging step** between initial generation and final editing. Practical demand is widespread: games require multiple style skins, product design needs style variants, and film animation requires stylized assets. Existing research focuses on generation or editing enhancement, leaving this intermediate step insufficiently addressed.
>
> Current methods (Paint3D, SyncMVD) rely on multi-view diffusion models, limited by data scarcity and computational cost. Our method performs iterative optimization on predefined or AI-generated UV maps, generating diverse results with fewer resources while alleviating partial-view missing and noise. This provides a concrete approach for addressing data scarcity and computational efficiency.
>
> We position this work as a practical method bridging generation and editing pipelines, incorporating technical innovations (LoRA enhancement, two-stage training, hybrid inference strategy) valuable for both industrial application and technical advancement.
>
> **W2: LVstyler performance is tightly coupled to CLIPstyler and inherits its limitations**
>
> LVstyler extends CLIPstyler with architectural and training improvements. Our LoRA-enhanced framework addresses three major problems when applying CLIPstyler to UV texture maps:
>
> 1. **High Noise:** CLIPstyler produces significant noise on UV maps. Our Shape LoRA, trained with Laplacian edge loss, reduces noise while preserving structural details (**Table 2**: PSNR 11.6884 vs. CLIPstyler's 10.7917).
>
> 2. **Over-stylization Breaking UV Region Independence:** CLIPstyler over-stylizes regions, breaking UV map region independence. Our two-stage training with UV LoRA adapts style while maintaining surface smoothness and region boundaries (**Figure 7**, **Table 2**: SSIM 0.4902 vs. CLIPstyler's 0.38875).
>
> 3. **Image Space Only:** CLIPstyler operates only in image space and cannot handle UV space characteristics (discontinuities, seams, non-uniform sampling). Our UV LoRA learns domain-specific adaptation, bridging image space to UV space.
>
> Our **contributions** are the dual LoRA architecture (Shape LoRA + UV LoRA) and two-stage training that extend CLIPstyler to 3D texture stylization, addressing domain-specific challenges. We will supplement the appendix with detailed CLIPstyler vs. LVstyler comparisons, demonstrating improvements in noise reduction and region independence preservation.
>
> **W3: CLIP-based text guidance is a bottleneck for abstract or culturally specific prompts**
>
> CLIP text guidance has known limitations for abstract or culturally specific prompts. **CLIP is only one part of our overall objective**: unlike CLIPstyler which relies solely on CLIP supervision, our loss function combines CLIP losses, content loss (VGG-based), total variation regularization, and domain-specific losses (Laplacian edge loss, L1 loss). This multi-objective design reduces over-reliance on CLIP.
>
> - **Dual LoRA Architecture:** Shape LoRA enforces geometric constraints independently of CLIP, while UV LoRA adapts style to UV space characteristics.
>
> - **Hybrid Inference Strategy:** We combine frozen LoRA modules (offline-learned constraints) with online ISS optimization, maintaining geometric constraints independent of CLIP limitations.
>
> - **UV Space Advantages:** Operating directly in UV space avoids complex spatial reasoning, reducing the 3D reasoning demands that CLIP struggles with.
>
> Similar to peer methods (Paint3D, TEXTure, Text2Tex), abstract and culturally-specific styles remain challenging. Our method handles common artistic styles well (**Figure 4**, **Table 1**: CLIP score 0.9161, Aesthetic score 4.3717), but highly abstract or culturally-specific prompts may still fail. We will consider addressing these cases in future work.
>
> **W4: Many loss weights introduced without hyperparameter ablation or sensitivity analysis**
>
> Our primary innovations are the dual LoRA architecture (Shape LoRA + UV LoRA) and hybrid inference strategy (weight injection from ISS to LVNet). Our ablation studies focus on:
>
> - The effectiveness of adding each LoRA module (**Table 2**, **Figures 6 and 7**)
> - The optimal timing of weight injection during optimization (**Figures 5 and 6**)
>
> Loss weights were selected based on empirical tuning and prior work, maintaining generalization capability. While hyperparameter sensitivity analysis would be valuable, our core innovations lie in architectural design and training strategy rather than hyperparameter optimization. Empirical evaluation demonstrates robust performance with selected weights across diverse test cases.

---

> ### Author Response · Authors · 2025-11-26
> **Thank you for your questions regarding comparison to feed-forward diffusion approaches, behavior under semantically distant prompts, and the necessity of our two-stage training scheme.**
>
> **Q1: Comparison to a feed-forward diffusion approach**
> Our optimization-based approach has advantages over feed-forward fine-tuning. The proposed solution is similar to Paint3D: training a UV diffusion model on large datasets. As shown in **Figure A.2**, even after fine-tuning on high-quality data, pre-trained models exhibit artifacts and missing regions, especially for unseen objects and views (OOD cases), whereas our optimization-based scheme adapts per object and avoids this degradation.
>
> 1. **Domain Knowledge Storage in LoRA:** We store UV space domain and shape knowledge in pre-trained LoRA blocks, capturing fundamental structural information while remaining lightweight and adaptable.
> 2. **Real-time Style Adjustment:** Unlike fixed feed-forward models, our optimization-based approach allows real-time style adjustment. Users control stylization degree by adjusting optimization iterations.
> 3. **Superior Quality and Flexibility:** Our method outperforms diffusion fine-tuning: **Quality:** Combining pre-trained LoRA constraints with online optimization avoids OOD performance degradation (**Table 1**: FID 191.05 vs. Paint3D's 195.87, LPIPS 0.1004 vs. Paint3D's 0.1027). **Flexibility:** Optimization-based methods support per-prompt adaptation, adjusting in real-time to specific style requirements, whereas fine-tuned models are fixed after training.
>
> **Why Optimization-Based Methods Are Better Suited:**
>
> Re-stylization requires handling diverse objects and styles not adequately represented in training data. While feed-forward fine-tuning may seem direct, existing implementations like Paint3D suffer from artifacts and OOD issues. Optimization-based methods adapt to each case through online learning, while feed-forward models are constrained by their training distribution. Our hybrid approach combines pre-trained LoRA modules with online optimization, providing superior quality and flexibility through LoRA-stored domain knowledge and real-time adjustment.
>
> **Q2: Behaviour of the method under semantically very distant styling prompts**
>
> Regarding semantically distant style prompts, CLIP may struggle with highly abstract or culturally-specific concepts outside its training distribution. However, our method handles certain culturally-specific and abstract styles. For example:
>
> - In **Figure A.5**, our prompt is "decorative bottle with the Great Wave off Kanagawa by Hokusai," and our method successfully reproduces this style.
> - In our tests, many abstract styles such as Van Gogh style can also be effectively reproduced.
> - Similar to your suggestion, our example in **Figure A.4** applies fire element style to a goldfish "a goldfish with fire skin", achieving reasonably successful results compared to peers.
>
> However, long prompts containing 20+ conditions are ineffective for our method, which focuses on re-stylization rather than full-scene painting and cannot simultaneously handle so many style and object elements.
>
> **Task Definition and Scope:**
> For re-stylization, we change the style while maintaining the 3D shape. Our method handles any shape as an independent re-stylization target. Shape editing is a different task. However, we can generate effects such as "a cart with flowing water and light on it" as a stylization of the cart surface texture, clarifying our scope for texture re-stylization tasks. The output can be seen in **Figure A.1**.
>
> **Q3: Necessity of the two-stage scheme versus a simpler joint-training variant**
>
> Joint training optimizes Shape LoRA and UV LoRA simultaneously but leads to suboptimal results:
>
> 1. **Conflicting Objectives:** Shape LoRA preserves structural edges while UV LoRA adapts style. Joint training causes these to compete, leading to unstable optimization.
>
> 2. **Loss Function Conflicts:** Stage 1 emphasizes shape preservation (λ_Lap = 50.0, λ_dir = 500.0), while Stage 2 reduces constraints (λ_c = 0.1, λ_dir = 50.0) for stylization. Joint training cannot balance these requirements.
>
> 3. **Training Instability:** Without a stable Shape LoRA foundation, UV LoRA struggles to learn UV adaptation, producing artifacts or over-stylization.
>
> The sequential two-stage process is critical:
>
> 1. **Separation of Concerns:** Stage 1 establishes shape preservation. Once Shape LoRA is frozen, Stage 2 focuses on UV adaptation without compromising structure.
>
> 2. **Stable Optimization:** Freezing Shape LoRA in Stage 2 provides stable constraints for UV LoRA, preventing deviation from shape-preserving solutions.
>
> 3. **Experimental Evidence:** Ablation study shows progressive improvements: Stage 1 achieves PSNR 11.6884 and SSIM 0.4440, Stage 2 achieves SSIM 0.4902, indicating better trade-offs.
>
> 4. **Modular Design:** Two-stage enables hybrid inference with frozen LoRA constraints and online ISS adaptation. Joint training cannot achieve this due to tight coupling.
>
> Experiments demonstrate that sequential two-stage training is critical for each LoRA module to learn its role without interference.

---

### Official Review · Reviewer_TpZN · 2025-11-04

**Soundness:** 3
**Presentation:** 3
**Contribution:** 3
**Rating:** 6
**Confidence:** 4

**Summary:**

The paper introduces LVstyler, a two-stage, LoRA-enhanced framework for UV texture stylization on 3D meshes. It first performs CLIP-guided online learning in the image space (ISS) to acquire style weights, then injects them into a lightweight network equipped with Shape LoRA for edge/structure preservation and UV LoRA for UV-space adaptation, enabling varied, text-driven styles while maintaining geometric fidelity. Experiments on Objaverse/ModelNet40 with a curated UV dataset report improved FID/LPIPS and Aesthetic scores, competitive CLIP alignment, and markedly lower GFLOPs and inference time versus Latent-Paint, TEXTure, Text2Tex, and Paint3D; user/VLM studies further support the perceived quality.

**Strengths:**

S1. Operates directly in UV space with LoRA constraints, reducing reliance on heavy diffusion back-projection and improving efficiency.

S2. The split of Shape LoRA and UV LoRA is well-motivated, offering a clear mechanism to balance structure preservation and stylistic diversity.

S3. Provides a practical pipeline and dataset for industry-style workflows, with strong runtime and compute advantages alongside competitive quality.

**Weaknesses:**

W1. The method appears to presuppose clean, fixed UV mapping and to operate purely in texture space. Any substantive geometric modification would typically invalidate the original UV parameterization and thus the training and inference assumptions. Please clarify whether the approach is strictly a stylization pipeline or if you have a mechanism to remain valid under geometry changes.

W2. I wonder how the authors obtain the GT data for stylization. Please clarify what you consider “ground truth,” how it is produced or curated, and whether it is true pixel-level GT or pseudo-GT derived from renderings.

W3. Please explain why PSNR decreases after the second stage, as shown in Table 2. Specify the reference used for PSNR (original texture vs. stylized target vs. rendered views) and discuss the trade-off between style strength and reconstruction fidelity.

**Questions:**

Please see the weakness.

---

> ### Author Response · Authors · 2025-11-26
> **Thank you for the detailed and constructive feedback!**
>
> **W1: Assumes clean, fixed UV mapping; unclear robustness under geometry changes**
>
> The original UV maps are obtained by projecting any 3D object surface onto UV space and may be either artist-designed or AI-generated; our method **does not presuppose clean, fixed UV mapping**. In our experiments, most inference is conducted on **coarse AI-generated UV maps** containing artifacts, missing angles, and high noise, yet LVstyler still produces diverse styles, complete coverage, and reduced noise. This empirically shows that we do not require pre-cleaned UV mappings: **different geometric variations do not affect the validity of our method**, since different 3D surfaces can all be projected to UV maps and handled uniformly by our framework.
>
> We believe this concern mainly arises from an implicit assumption that our method depends on pre-defined, artifact-free UV maps. To avoid this misunderstanding, we will in the revision:
> - Add clearer descriptions of input requirements, explicitly stating that our method accepts UV maps from various sources, including problematic AI-generated UV maps, and provide representative input examples in the appendix (**Figure A.1**) showing that our method remains effective under UV maps of varying quality and different geometric configurations, without requiring fixed UV parameterization.
>
> **W2: Definition and construction of stylized "ground truth" data**
>
> This concern is closely related to how we construct supervision signals for training and what we regard as "ground truth." Our method **does not rely on a large corpus of manually stylized, pixel-level ground-truth UV textures**; instead, we treat the **original UV map as the content/structure reference** and use a three-part decoupled design to provide stable supervisory signals while avoiding heavy GT annotation.
>
> Our method decouples 3D object stylization into three independent components:
>
> 1. **Style Learning:** We employ CLIP-based contrastive learning on LVNet outputs to enhance style consistency. This component focuses on learning the target style from text prompts, independent of UV map structure.
>
> 2. **UV Map Structure Preservation:** We only need to perform edge detection on the original UV maps using operators such as Sobel or Prewitt. This minimal preprocessing requirement significantly reduces the dependency on clean, fixed UV mappings. Edge detection provides sufficient structural information for our Shape LoRA to preserve geometric details without requiring perfect UV parameterization.
>
> 3. **UV Map Domain Consistency:** We perform structural alignment between the stylized UV map output and the original UV map. This ensures that while stylizing, we maintain the isolation of object parts (e.g., eyes and hands do not blend together), preserving the independence of different UV map regions.
>
> Through this design, our training no longer requires large quantities of complete, clean UV maps, yet still outputs consistent and high-quality UV maps, addressing two fundamental challenges:
>
> - **Data Scarcity:** We mitigate limited training data by not requiring perfect UV mappings.
> - **Artifact Reduction:** We reduce diffusion artifacts by operating directly in UV space with explicit structural constraints, rather than relying on multi-view diffusion with accumulating errors.
>
> Consequently, this three-part decoupling makes our method robust to geometric changes. Since we only require edge detection (applicable to any UV map projection) and structural alignment (adapting to input UV structure), our method can handle UV maps from various sources and geometric configurations without assuming fixed parameterization.
>
> **W3: PSNR trade-off after Stage 2**
>
> We compute **PSNR** = 20log10(255) − 10log10(MSE) per-view with the **original UV map as reference**, measuring per-pixel differences between generated and reference images. The PSNR decrease stems from Stage 2 actively amplifying **style intensity**. The UV LoRA projects visual style more strongly onto UV space, deliberately **deviating from the original UV texture** to reinforce style details. This increases pixel-level differences from the reference, naturally lowering PSNR (per-pixel reconstruction fidelity). However, Stage 2 achieves improved **SSIM** of 0.4902 and maintains **LPIPS** at 0.7359 (**Table 2**), indicating better structural consistency and smaller perceptual differences. The rendered results receive higher visual quality scores, demonstrating an explicit **trade-off between stronger style expression and pixel-wise reconstruction fidelity**—the deviation yields style expression more aligned with high-quality perceptual standards.

---

### Note · Program_Chairs · 2026-01-17
**Submission Desk Rejected by Program Chairs**

The following references in this submission do not refer to real documents and/or have major errors in bibliographic information:

 Gideon Reich, Sequoia Nutshell, and Lior Wolf. Text2tex: Controllable 3D texture synthesis using text-to-image diffusion. arXiv preprint arXiv:2310.03641, 2023.
Yonghwan Ryu, Songhyun Yu, Jeyoung Seo, Yunmo Bae, and Sunghyun Cho. CLIPAG: Robust vision-language models with perceptually aligned gradients. In IEEE/CVF International Conference on Computer Vision (ICCV), 2023.
Yongxin Huang, Zhizhong Han, and Yu-Shen Liu. Beziersketch: Stroke-based artistic style transfer. ACM Transactions on Graphics (TOG), 40(4):60:1-60:13, 2021.
Hyungtaek Kim, Minuk Ma, Seoung Wug Oh, and Seungjin Choi. Training-free style injection in diffusion models. In IEEE/CVF International Conference on Computer Vision (ICCV), 2023.
Shuyi Wang, Kai Han, and Tong Lu. U-stydit: Transformer meets u-net for high-resolution style diffusion. In European Conference on Computer Vision (ECCV), 2024.
Jiashen Wu, Chenlin Ma, Xiaowei Tang, and Wen Jiang. DiffLoRA: Zero-shot personalization via latent-diffusion hyper-networks. arXiv preprint arXiv:2403.08211, 2024.